



# A double peak in the seasonality of California's photosynthesis as observed from space

Alexander J. Turner[1,2,3], Philipp Köhler[4], Troy S. Magney[3,4,5], Christian Frankenberg[3,4], Inez Fung[1], and Ronald C. Cohen[1,2]

[1]Department of Earth and Planetary Sciences, University of California, Berkeley, CA, 94720, USA.
[2]College of Chemistry, University of California, Berkeley, CA, 94720, USA.
[3]Jet Propulsion Laboratory, California Institute of Technology, Pasadena, CA, 91109, USA.
[4]Division of Geological and Planetary Sciences, California Institute of Technology, Pasadena, CA, 91226, USA.
[5]Department of Plant Sciences, University of California, Davis, CA, 95616, USA.

**Correspondence:** Alexander J. Turner (alexjturner@berkeley.edu)

**Abstract.** Solar-Induced chlorophyll Fluorescence (SIF) has been shown to be a powerful proxy for photosynthesis and gross
primary productivity (GPP). The recently launched TROPOspheric Monitoring Instrument (TROPOMI) features the required
spectral resolution and signal-to-noise ratio to retrieve SIF from space. Here we present an oversampling and downscaling
method to obtain 500-m spatial resolution SIF over California. We report daily values based on a 14-day window. TROPOMI
SIF data show a strong correspondence with daily GPP estimates at AmeriFlux sites across multiple ecosystems in California.
We find a linear relationship between SIF and GPP that is largely invariant across ecosystems with an intercept that is not
significantly different from zero. Measurements of SIF from TROPOMI agree with MODIS vegetation indices (NDVI, EVI,
and $NIR_v$) at annual timescales but indicate different temporal dynamics at monthly and daily timescales. TROPOMI SIF
data show a double peak in the seasonality of photosynthesis, a feature that is not present in the MODIS vegetation indices.
The different seasonality in the vegetation indices may be due to a clear-sky bias in the vegetation indices, whereas SIF has
a low sensitivity to clouds and can detect the downregulation of photosynthesis even when plants appear green. We further
decompose the spatio-temporal patterns in the SIF data based on land cover. The double peak in the seasonality of California's
photosynthesis is due to two processes that are out of phase: grasses, chaparral, and oak savanna ecosystems show an April
maximum while evergreen forests peak in June. An empirical orthogonal function (EOF) analysis corroborates the phase offset
and spatial patterns driving the double peak. The EOF analysis further indicates that two spatio-temporal patterns explain 84%
of the variability in the SIF data. Results shown here are promising for obtaining global GPP at sub-kilometer spatial scales
and identifying the processes driving carbon uptake.

## 1 Introduction

Photosynthesis is the process by which plants and other organisms use sunlight to synthesize carbon dioxide ($CO_2$) and water
to glucose and oxygen. Accurate knowledge of gross primary productivity (GPP) through photosynthesis is crucial for under-
standing the land-atmosphere carbon exchange, which is one of the largest and most uncertain aspects of the global carbon





cycle (IPCC, 2013; Anav et al., 2015; USGCRP, 2018). This uncertainty in the land-atmosphere carbon exchange has led to
long-standing questions regarding the magnitude of the Northern Hemispheric terrestrial carbon sink and how it has changed
over the past few decades (e.g., Tans et al., 1990; Ballantyne et al., 2012; Ciais et al., 2019). As such, more direct methods of
inferring GPP are of great interest to the scientific community.
Previous work estimating regional or global scale GPP has typically relied on either biosphere models (e.g., the early work
on SiB2 from Sellers et al., 1986), used remote-sensing measurements in Monteith light use efficiency models with scalings
for different ecosystems and climatic conditions (e.g., Monteith, 1972; Mahadevan et al., 2008), or attempted to back out GPP
from $CO_2$ flux inversions (e.g., CarbonTracker from Peters et al., 2007). The advent of global remote-sensing observations
of Solar-Induced chlorophyll Fluorescence (SIF) represents a breakthrough in our ability to constrain photosynthetic activity
from space. This is because a number of studies have shown SIF to be a powerful proxy for photosynthesis both in laboratory
environments (e.g., Baker, 2008) and at larger spatial scales (e.g., Frankenberg et al., 2011a; Parazoo et al., 2014; Yang et al.,
2015, 2017; Sun et al., 2017, 2018b; Magney et al., 2019a). During the initial stage of photosynthesis, absorbed sunlight excites
chlorophyll $a$ molecules. The primary pathways for de-excitation are via photochemistry or non-photochemical quenching, the
latter of which dissipates excess energy as heat when the plant does not have the capacity for photosynthesis (i.e. under stress).
However, a small fraction dissipates as heat or is re-emitted as fluorescence and can be measured by remote sensing. This
remote sensing retrieval is termed SIF.
The first global space-borne measurements of SIF were made by Frankenberg et al. (2011b) and Joiner et al. (2011) using
observations from the Japanese GOSAT instrument (Kuze et al., 2009). Since then, SIF has been retrieved from other space-
borne instruments such as: GOME-2 on the METOP-A satellite, SCIAMACHY on the ENVISAT satellite, the OCO-2 satellite,
and TROPOMI on the Sentinel-5 Precursor satellite (Frankenberg et al., 2011a, b, 2012, 2014; Joiner et al., 2011, 2012,
2013, 2014, 2016; Guanter et al., 2012, 2015; Köhler et al., 2015, 2018). A number of upcoming satellite missions such
as FLEX (Drusch et al., 2017) and TEMPO (Zoogman et al., 2017) will also measure SIF at higher spatial and temporal
resolution. Efforts are underway to create a multi-decadal SIF record using different space-borne instruments (Parazoo et al.,
2019) and a few groups have utilized machine learning techniques to create spatially continuous SIF datasets at $0.05° \times 0.05°$
resolution (Zhang et al., 2018; Yu et al., 2019; Li and Xiao, 2019). Mohammed et al. (2019) presents a detailed review of
different remote sensing techniques for retrieving SIF from space-borne measurements.
Some work has shown SIF to be a better measure of carbon uptake than other vegetation indices that look at canopy "green-
ness". This is, in part, because indices like the normalized difference vegetation index (NDVI) are a measure of photosynthetic
capacity (Sellers et al., 1986) whereas SIF is a measure of the photosynthetic activity and is coupled to the radiation regime.
For example, Luus et al. (2017) showed that the seasonal cycle of a biosphere model driven by SIF agreed with measurements
of $CO_2$ whereas the seasonal cycle from the model driven by the Enhanced Vegetation Index (EVI) was markedly different
from the $CO_2$ observations. Joiner et al. (2011) showed that the seasonal cycles of SIF and EVI agree in some regions, but not
others. Walther et al. (2016) showed a decoupling of the photosynthesis and greenness dynamics in boreal evergreen forests
by comparing SIF and EVI to model estimates of GPP, with SIF better capturing the seasonality of both deciduous broadleaf
and evergreen needleleaf forests. Again, this is likely due to SIF capturing photosynthetic activity, rather than photosynthetic





capacity. More recently, Magney et al. (2019a) demonstrated a mechanistic link between SIF and GPP over the course of a
year in a winter-dormant Northern Hemisphere conifer forest, despite retaining chlorophyll through the winter. Magney et al.
(2019a) highlighted the potential for new satellite measurements of SIF from TROPOMI and OCO-2 to track GPP at coarse
spatial resolution ($3.5 \times 7$ km$^2$).

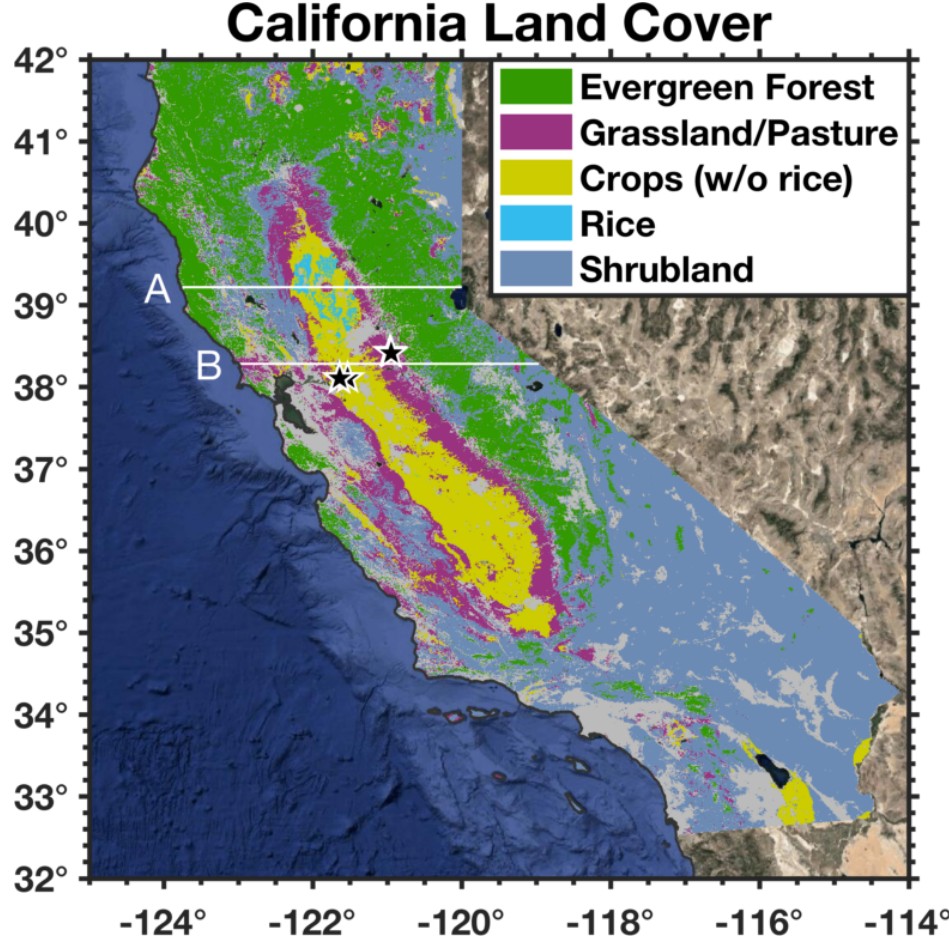

**Figure 1. California land cover.** Land cover in California from the 2018 USDA CropScape database (USDA, 2018). Resolution has been degraded from the native 30-m resolution to 500-m for comparison with TROPOMI data. Coloring indicates that a land type makes up more than 50% of the 500-m grid cell. White lines are the locations of two transects across California for Hovmöller diagrams: 39.218°N (Transect A), and 38.282°N (Transect B). Black stars show the location of six AmeriFlux sites: Bouldin Island (38.1090°N, 121.5350°W; US-Bi2), Tonzi Ranch (38.4316°N, 120.9660°W; US-Ton), Vaira Ranch (38.4133°N, 120.9507°W; US-Var), Twitchell Island West Pond (38.1074°N, 121.6469°W; US-Tw1), Twitchell Island East End (38.1030°N, 121.6414°W; US-Tw4), and Twitchell Island East Pond (38.1072°N, 121.6426°W; US-Tw5).





Here we present an oversampling and downscaling method to obtain daily estimates of SIF at 500-m resolution. To our
knowledge, this is the highest resolution SIF dataset from satellite measurements. We then compare this down-scaled 500-m SIF
data to AmeriFlux sites across the state of California to assess the relationship between SIF and GPP. We finish by decomposing
California's spatio-temporal patterns of photosynthesis and carbon uptake into the dominant modes using empirical orthogonal
functions (EOFs). Here we focus on California because there are a number of eddy flux towers and it encompasses a range of
diverse ecosystems including: deciduous and evergreen forests, irrigated croplands, and grasslands (see Fig. 1).
**2   Measurements of SIF, vegetation, and GPP**
**2.1   Satellite measurements of Solar-Induced chlorophyll Fluorescence (SIF) from TROPOMI**
The TROPOspheric Monitoring Instrument (TROPOMI; Veefkind et al., 2012) is a nadir-viewing imaging spectrometer with
bands in the UV, visible, shortwave infrared, and near-infrared aboard the Sentinel-5 Precursor satellite. The Sentinel-5 Pre-
cursor satellite was launched into low earth orbit on October 13, 2017 with an equatorial crossing time of 13:30 local solar
time and a 16 day orbit cycle. TROPOMI has a wide swath (2,600 km across track) enabling near-daily global coverage. The
spatial resolution of the ground pixels is 7 km along track and 3.5-15 km across track (3.5 km at nadir and 15 km at the edge of
the swath). Of particular relevance here is the near-infrared band (725-775 nm) that covers the far-red part of SIF emission and
contains a number of solar absorption features in the solar irradiance (Fraunhofer lines), allowing for retrieval of SIF through
the change in optical depth of Fraunhofer lines. Guanter et al. (2015) showed the potential for TROPOMI to retrieve SIF and
Köhler et al. (2018) presented the first retrievals. Specifically, Köhler et al. (2018) used a 743-758 nm retrieval window that
is devoid of atmospheric absorption features. TROPOMI has a spectral resolution of ∼0.4 nm and a signal-to-noise ratio of
∼2,500 in this retrieval window. The TROPOMI SIF retrieval uses a singular value decomposition to derive the spectral basis
functions from TROPOMI data over vegetation-free areas (e.g., oceans, ice, and deserts).
One particularly attractive feature of space-borne SIF retrievals is the low sensitivity to atmospheric scattering by aerosols
and clouds. Specifically, Frankenberg et al. (2012) showed that 80% of the emitted SIF could be retrieved in the presence of
clouds with low-to-moderate optical thickness. This weak sensitivity to clouds is in contrast to reflectance based measures of
vegetation (e.g., NDVI) that can only be made in clear-sky conditions, potentially inducing a clear-sky bias in reflectance based
vegetation indices.
Here, we apply one additional bias correction to the TROPOMI retrievals that was not included in Köhler et al. (2018). We
find some mostly barren regions have systematically negative SIF values, which is non-physical. This bias is thought to be
related to bright surfaces and is likely due to the choice of training data for the spectral basis functions. We are investigating
ways to correct this globally. In the interim, we compute a spatio-temporal bias correction $b_{i,j,k}$ (where $i,j$ are the spatial
coordinates and $k$ is the temporal coordinate) such that the mean SIF for a given location over a 30-day moving window is





always positive. That is to say,

$$
b_{i,j,k} = \begin{cases} |\bar{s}_{i,j,k}|, & \bar{s}_{i,j,k} < 0 \\ 0, & 0 \leq \bar{s}_{i,j,k} \end{cases}
\tag{1}
$$

where $\bar{s}_{i,j,k}$ is the 1-month block average for the $k^{\text{th}}$ day at location $i, j$. This still allows for negative SIF values due to vari-
ability and noise but will shift the mean SIF for a given 500-m grid cell to be positive. In practice, this bias correction is small
with 78% having no bias correction at all and 89% of the grid cells having a bias correction smaller than 0.1 mW/m$^2$/sr/nm.
The bias correction primarily impacts desert regions in Southeastern California (see Supplemental Fig. S1).
## 2.2 Satellite-based vegetation indices from MODIS
MODerate Resolution Imaging Spectroradiometer (MODIS) is an imaging spectrometer on NASA's Terra and Aqua satellites.
Terra launched in 2000 and Aqua launched in 2002, both are in sun-synchronous orbits with 16 orbits per day. Terra and Aqua
have equatorial crossing at 10:30 and 12:00 local solar time, respectively. Schaaf et al. (2002) developed the Nadir Bidirectional
reflectance distribution function-Adjusted Reflectance (NBAR) dataset, hereafter referred to as the MODIS NBAR dataset.
MODIS data over a 16-day period from Terra and Aqua can be combined to build a 500-m composite: MCD43A4. Here
we use the MCD43A4.006 (v06) MODIS NBAR dataset to compute three MODIS vegetation indices at 500-m resolution.
Specifically, we compute the normalized difference vegetation index (NDVI), enhanced vegetation index (EVI), and near-
infrared reflectance of vegetation index (NIR$_{\text{v}}$):

$$
\text{NDVI} = \frac{\rho_{\text{NIR}} - \rho_{\text{red}}}{\rho_{\text{NIR}} + \rho_{\text{red}}}
\tag{2}
$$

$$
\text{EVI} = G \cdot \frac{\rho_{\text{NIR}} - \rho_{\text{red}}}{\rho_{\text{NIR}} + C_1 \rho_{\text{red}} - C_2 \rho_{\text{blue}} + L}
\tag{3}
$$

$$
\text{NIR}_{\text{v}} = \rho_{\text{NIR}} \cdot \text{NDVI}
\tag{4}
$$

where $\rho_{\text{NIR}}$, $\rho_{\text{red}}$, and $\rho_{\text{Blue}}$ are the reflectances in their respective MODIS bands and $G$, $C_1$, $C_2$, and $L$ are coefficients for the
MODIS EVI algorithm ($L = 1$, $C_1 = 6$, and $C_2 = 7.5$, $G = 2.5$).
## 2.3 GPP estimates from AmeriFlux eddy covariance sites
AmeriFlux is a network of long-term eddy covariance sites that launched in 1996 (Baldocchi et al., 2001). These eddy covari-
ance sites provide a direct measure of net ecosystem exchange (NEE; CO$_2$ fluxes) (Baldocchi et al., 1988) and can be used
to evaluate both bottom-up models and satellite proxies of carbon exchange. Disentangling the CO$_2$ fluxes into GPP (CO$_2$
uptake) and total ecosystem respiration ($R_{\text{eco}}$; CO$_2$ released) generally requires making assumptions about the temperature
dependence of the respiration which can induce biases in the GPP estimate (Reichstein et al., 2005). Nevertheless, these eddy
covariance sites provide the best estimate of site level GPP across multiple ecosystems in California including: croplands, wet-
lands, woody savannas, and grasslands. Here we use data from 11 AmeriFlux sites across California (see Table 1) to evaluate
the SIF retrievals from TROPOMI. NEE flux partitioning at these sites was performed using artificial neural networks from
nighttime measurements to constrain $R_{\text{eco}}$ (Hemes et al., 2019).





**Table 1.** AmeriFlux sites used in this study.

| Site ID | Site name | Latitude (°N) | Longitude (°W) | Elevation (m a.s.l.) | Vegetation type[a] |
|---------|-----------|---------------|----------------|----------------------|--------------------|
| US-Bi1 | Bouldin Island Alfalfa | 38.0992 | 121.4993 | -3 | CRO[b] |
| US-Bi2 | Bouldin Island Corn | 38.1090 | 121.5350 | -5 | CRO[b] |
| US-EDN | Eden Landing Ecological Reserve | 37.6156 | 122.1140 | 1 | WET[c] |
| US-Myb | Mayberry Wetland | 38.0499 | 121.7650 | -4 | WET[c] |
| US-Sne | Sherman Island Restored Wetland | 38.0369 | 121.7547 | -5 | GRA[d] |
| US-Ton | Tonzi Ranch | 38.4316 | 120.9660 | 177 | WSA[e] |
| US-Tw1 | Twitchell Island West Pond | 38.1074 | 121.6469 | -9 | WET[c] |
| US-Tw3 | Twitchell Island Alfalfa | 38.1159 | 121.6467 | -9 | CRO[b] |
| US-Tw4 | Twitchell Island East End | 38.1030 | 121.6414 | -5 | WET[c] |
| US-Tw5 | Twitchell Island East Pond | 38.1072 | 121.6426 | -5 | WET[c] |
| US-Var | Vaira Ranch | 38.4133 | 120.9507 | 129 | GRA[d] |

[a] Vegetation classification based on the International Geosphere-Biosphere Programme (IGBP) classification scheme (Strahler et al., 1999).

[b] CRO (Croplands): Lands covered with temporary crops followed by harvest and a bare soil period (e.g., single and multiple cropping systems). Note that perennial woody crops will be classified as the appropriate forest or shrub land cover type.

[c] WET (Permanent Wetlands): Lands with a permanent mixture of water and herbaceous or woody vegetation that cover extensive areas. The vegetation can be present in either salt, brackish, or fresh water.

[d] GRA (Grasslands): Lands with herbaceous types of cover. Tree and shrub cover is less than 10%.

[e] WSA (Woody Savannas): Lands with herbaceous and other understory systems, and with forest canopy cover between 30-60%. The forest cover height exceeds 2 meters.

## 2.4 Comparison of TROPOMI SIF with MODIS vegetation indices

Figure 2 shows a scatterplot comparison of TROPOMI SIF and MODIS NDVI, EVI, and $NIR_v$. The comparison is limited to coincident observations between March and August (MAMJJA) and excludes scenes that are predominantly barren or shrubland. A few features that immediately stand out are:

1. The strong correspondence between EVI and $NIR_v$. We find a nearly linear relationship between these two indices ($r^2 = 0.98$).

2. All three MODIS indices are well correlated with eachother ($r^2 > 0.85$). We do observe a weakly non-linear relationship between $NIR_v$ and EVI or NDVI (see the curvature in the $NIR_v$ row).



**Figure 2. Comparison of MODIS vegetation indices and TROPOMI SIF from 2018-2019.** Panels show a comparison of coincident measurements in both space and time of NDVI, EVI, NIR$_v$, and SIF. NDVI, EVI, and NIR$_v$ use the 500-m MODIS BRDF-corrected reflectances and SIF is from TROPOMI. Shading indicates the density of points. Data is filtered to only include measurements from March through August (MAMJJA). Data is further filtered to remove scenes that are more than 85% barren or shrubland as defined by the CropScape Database. Gray histogram on the x/y-axes show the distribution of values for a given set of data. Supplemental Fig. S2 shows the comparison including the SIF downscaled using local MODIS vegetation indices.





3. The weaker relationship between SIF and the vegetation indices. Previous work has argued that $NIR_v$ is strongly cor-
related with SIF (Badgley et al., 2017) and provides a new independent approach for estimating GPP (Badgley et al.,

3      2019).

Of the three vegetation indices examined here, we find the strongest relationship between $NIR_v$ and SIF, but it only explains
half of the variability on daily timescales ($r^2 = 0.52$). The agreement improves at coarser temporal scales (annual $r^2 = 0.83$–
0.84 and monthly $r^2 = 0.59 \pm 0.23$). It is important to note that the native spatial resolution of the TROPOMI observations
are 3.5-km across-track at nadir, whereas the MODIS observations are 500-m across-track at nadir. As such, we are using
all MODIS observations within a single TROPOMI scene. Comparison of four methods of downscaling SIF with $NIR_v$ yield
coefficients of determination of $r^2 = 0.52 - 0.64$; see Fig. S2).
**3    Oversampling and spatial-downscaling of TROPOMI data**

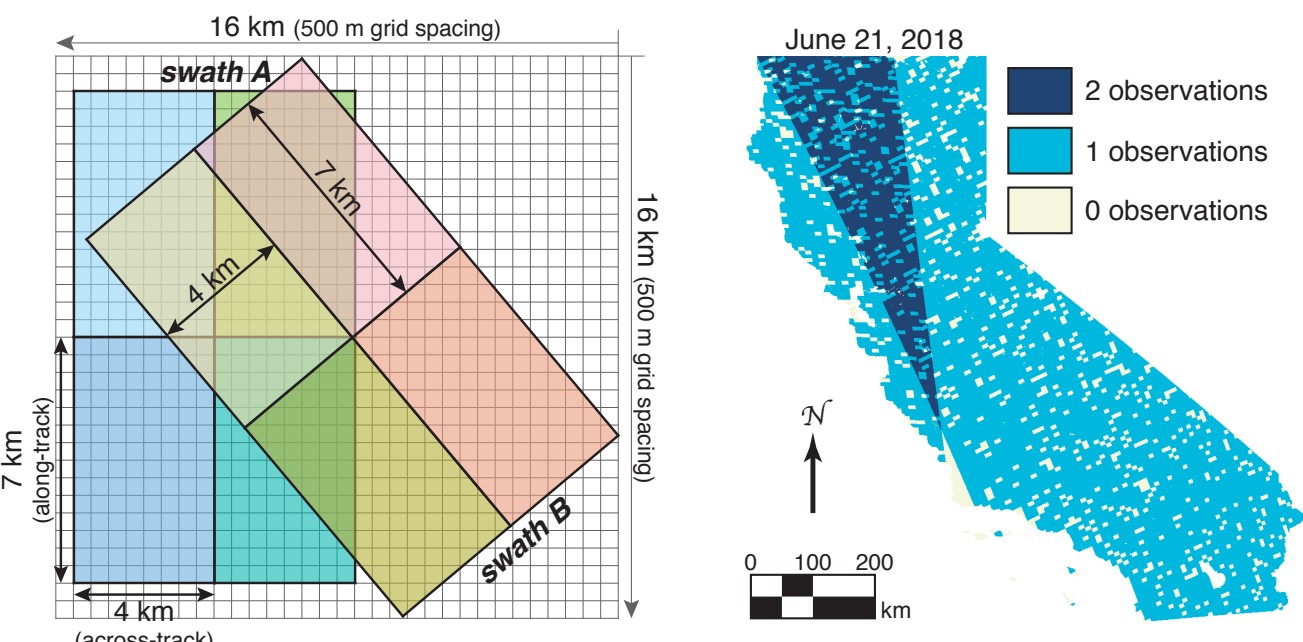

**Figure 3. Oversampling schematic.** Left panel shows the schematic for our oversampling. Gray grid has a grid spacing of 500 m (equivalent to the spatial resolution of the MODIS MCD43A4 product). TROPOMI ground pixels are 7 km along-track and vary from 3.5 km (at nadir) to 15 km off swath edge) across-track. Schematic shows the spatial extent of eight hypothetical TROPOMI scenes from two swaths at 7×4 km², individual TROPOMI scenes are a different color. Swath B is rotated 40° relative to swath A, resulting in overlapping pixels. Right panel shows the number of successful retrievals on June 21, 2018 over California plotted on a 500-m grid.

As mentioned above, the nominal spatial resolution of the ground pixels from TROPOMI are 3.5×7 km² at nadir. However,
the wide swath from TROPOMI (2600 km across-track) often results in multiple observations per day (see right panel of





Fig. 3). Additionally, the orientation of these swaths differ over the 16-day orbit cycle allowing us to infer higher spatial
resolution than the nominal spatial resolution. This idea has been widely used with the space-borne OMI instrument that
preceded TROPOMI (see Sun et al., 2018a, and references therein for a discussion of oversampling with OMI observations).
However, the spatial resolution of TROPOMI is a factor of 15 finer than OMI ($3.5 \times 7$ km$^2$ for TROPOMI and $14 \times 26$ km$^2$ for
OMI, both at nadir). Oversampling with OMI often required years of observations (e.g., Zhu et al., 2014). The wide swath and
high spatial dense spatial coverage of TROPOMI allow us to perform biweekly oversampling.
Fig. 3 shows a schematic of how the oversampling is performed. The left panel shows two hypothetical swaths from
TROPOMI overlaid on a 500-m grid (same spatial resolution as the MODIS NBAR dataset). Areas where the swaths overlap
allow for us to partition the information down to a finer spatial scale. For example, the yellow pixel in swath B overlaps with all
four pixels from swath A. As such, the signal from that pixel in swath B can be sub-divided to finer spatial scales. Each unique
shade of color would correspond to unique information in the left panel of Fig. 3. The right panel of Fig. 3 shows the sampling
density of TROPOMI over California on a single day in June 2018, the dark blue region indicates where two TROPOMI swaths
overlapped that day.
We find that, on average, each 500-m grid cell is within the bounds of ∼0.6 TROPOMI scenes with a successful retrieval per
day. By using biweekly oversampling (a moving 14-day window) we obtain approximately 8 different swath orientations over
a 14-day period for the oversampling. These 8 swath orientations allow us to further refine our grid to following the schematic
shown in Figure 3. It also means that the daily values presented here are representative of 14-day moving averages (centered
about that day).
We can take the oversampling a step further by pre-weighting the SIF signal in a TROPOMI scene by the underlying
vegetation fraction, we refer to this as "downscaling". That is to say, we assume the observed SIF from TROPOMI in a given
scene likely originates from more vegetated regions within that scene. Here we use a relative weighting for this downscaling:
$$s_{i,j} = s^\star \frac{v_{i,j}}{\bar{v}} \qquad (5)$$
where $s^\star$ is the retrieved SIF from TROPOMI for a single scene, $s_{i,j}$ is the SIF spatially downscaled to 500-m, $v_{i,j}$ are the
vegetation indices from MODIS that fall within the bounds of a single scene from TROPOMI (i.e., the gray boxes within a
TROPOMI box in the left panel of Fig. 3), and $\bar{v}$ is the mean vegetation index over a given TROPOMI scene. Using Eq. 5 with
$\mathbf{v} = [1, \ldots, 1]$ returns the unweighted oversampling result. Following this, $s_{i,j}$ will naturally revert to oversampling in regions
with homogeneous vegetation (as inferred by MODIS).
Figure 4 shows the 2018 annual mean SIF from TROPOMI from Köhler et al. (2018) at $0.05° \times 0.05°$ spatial resolution
and California's seasonal cycle at weekly temporal resolution (non-bias corrected). The middle and bottom rows of Figure 4
show the 2018 annual mean SIF and seasonal cycle using oversampling and spatially downscaled with NIR$_v$ from MODIS. All
three show consistent large-scale spatial patterns. We do, however, find significant differences between the results from Köhler
et al. (2018) and the oversampling or downscaling method over the San Francisco Bay Area where the complex topography
induces numerical artifacts such as high SIF values over water. We also point out that the Köhler et al. (2018) seasonal cycle
is at weekly temporal resolution whereas we obtain daily temporal resolution because of the 14-day moving window. The

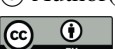

**Figure 4. 2018 annual mean photosynthesis.** Top row shows the 2018 annual mean TROPOMI SIF from Köhler et al. (2018) and inset shows the seasonal cycle (0.05° × 0.05° spatial resolution and weekly temporal resolution, respectively). Middle row uses the same TROPOMI SIF data but but oversampled to 500-m spatial resolution and daily temporal resolution. Bottom row uses an NIR$_v$-weighted local downscaling to 500-m spatial resolution. Left column shows all of California, USA and right column shows the San Francisco Bay Area in detail. Dashed black line in the left column indicates the domain of the right column.





oversampling and downscaling methods both yield consistent large-scale patterns and seasonal cycles (left panels in Fig. 4).
The main impact of the MODIS-based local downscaling is a sharpening effect. This can be seen in the right column of Fig. 4.
Importantly, the gradients observed in the oversampled SIF are also present in the downscaled SIF. The choice of MODIS
vegetation index to use in the downscaling makes little difference as the $r^2$ between the different downscaled SIF products
range from 0.99-1.00 (see Fig. S2), hereafter we use SIF downscaled with $NIR_v$ because, of the three vegetation indices, $NIR_v$
showed the strongest correlation to SIF (see Fig. 2). Again, we stress that the large-scale spatio-temporal patterns are conserved
between the oversampling and downscaling methods and the nuanced difference in processing allow for analysis at much finer
spatio-temporal scales. That is to say that we are not inducing large-scale changes in the spatio-temporal patterns with these
different methods of processing, those are robustly driven by the underlying SIF retrievals.
## 4  Inferring GPP from SIF
Previous work has shown strong empirical relationships between SIF and GPP at coarse spatial scales (e.g., Walther et al.,
2016; Jeong et al., 2017; Parazoo et al., 2018; Zuromski et al., 2018; Sun et al., 2018b). Magney et al. (2019a) recently
extended this SIF-GPP relationship by showing, in a conifer forest, how both SIF and GPP are regulated by seasonal changes
in photoprotective pigments and how SIF is directly related to needle physiology.
Lee et al. (2013), Guanter et al. (2014), and Sun et al. (2017) have previously argued for a linear relationship between SIF
and GPP, this follows from a simple relational analysis. From Monteith theory (Monteith, 1972) we can write:

$$\text{GPP} = \Phi_{\text{CO}_2}\alpha I \tag{6}$$

where $\Phi_{\text{CO}_2}$ is the light use efficiency of $CO_2$ assimilation, $I$ is the photosynthetically active radiation (PAR), and $\alpha$ is the
fractional absorbance of PAR. An analogous expression can be written for SIF (Lee et al., 2013):

$$\text{SIF} = \Phi_F \alpha \beta I \tag{7}$$

where $\Phi_F$ is the the light-use efficiency of SIF and $\beta$ is the probability of SIF photons escaping the canopy. Rearranging yields:

$$\text{GPP} = \frac{\Phi_{\text{CO}_2}}{\beta \Phi_{\text{F}}}\text{SIF}. \tag{8}$$

From Eq. 8 we can see that GPP should be proportional to SIF. However, there are likely differences in $\Phi_{\text{CO}_2}/(\beta \Phi_F)$ between
ecosystems. Notably, Yang et al. (2018) argued that SIF is more strongly correlated with the absorbed PAR ($\alpha I$) than with GPP
at sub-daily timescales, which implicitly points to non-linearities in $\Phi_{\text{CO}_2}/(\beta \Phi_F)$. $\beta$ will be a function of the canopy structure
and likely differs between ecosystems, although some studies have argued that reflectance measurements could be used to infer
$\beta$ (Yang and van der Tol, 2018; Zeng et al., 2019). Additionally, the ratio of $\Phi_{\text{CO}_2}$ to $\Phi_F$ will likely be ecosystem specific
due to, for example, differences in photosynthetic pathways (C3 versus C4 plants; Liu et al., 2017). A number of studies have
found the relationship between chlorophyll fluorescence and GPP to be non-linear at the leaf-scale (e.g., Magney et al., 2017,





2019b), owing the increased linearity at the canopy scale to averaging SIF and GPP over many different leaf angles exposed to
highly heterogenous light environments.

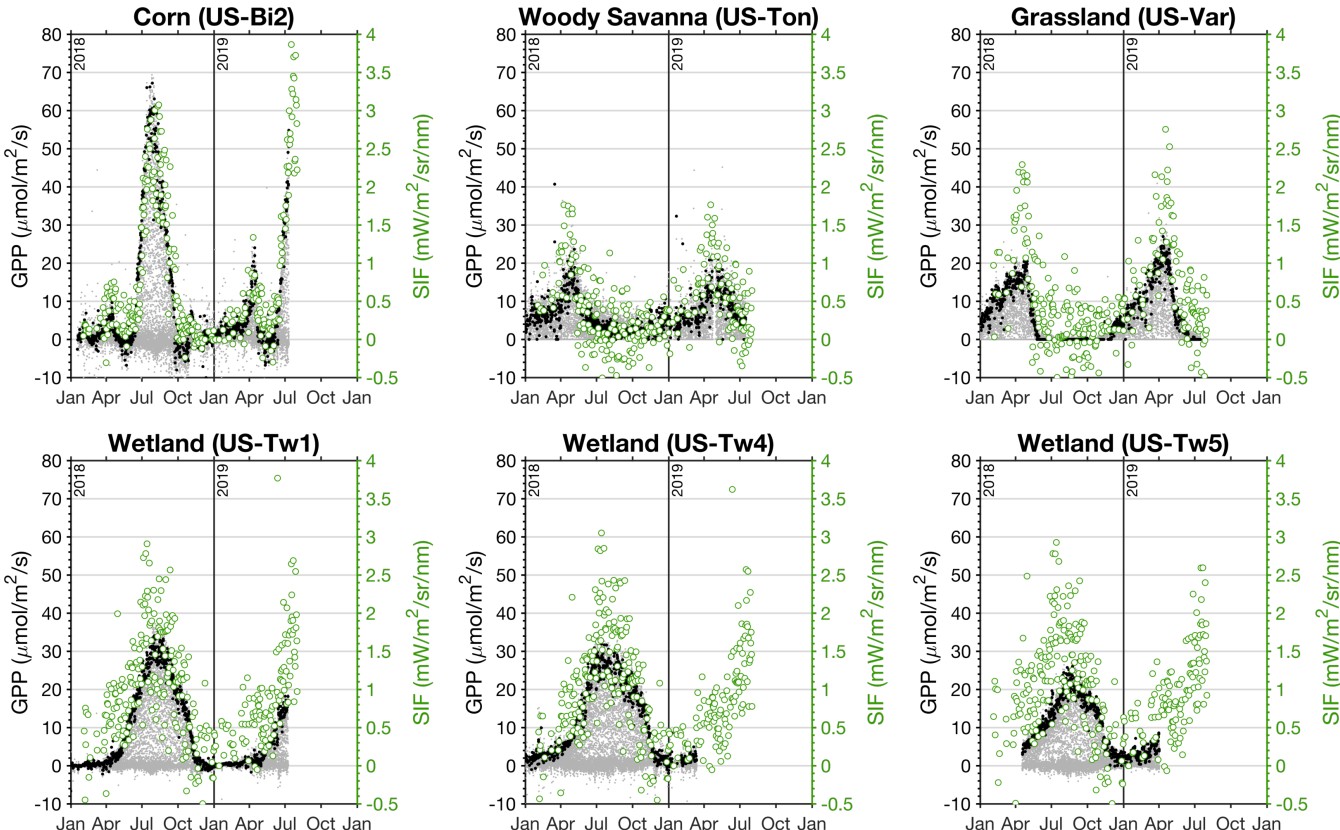

**Figure 5. AmeriFlux GPP and TROPOMI SIF at six sites in California.** Left axes (black) show GPP from AmeriFlux and right axes (green) show SIF from TROPOMI that have been downscaled with NIR$_v$. Light gray dots show all of the GPP measurements from the AmeriFlux site and black dots indicate GPP measurements between 13:00-14:00 PST (TROPOMI equatorial overpass is 13:30 local solar time). Green circles show the TROPOMI SIF observations at the AmeriFlux sites after applying the scene-specific relative weighting from the MODIS NIR$_v$. AmeriFlux sites used are: Bouldin Island (US-Bi2; 38.1090°N, 121.5350°W), Tonzi Ranch (US-Ton; 38.4316°N, 120.9660°W), Vaira Ranch (US-Var; 38.4133°N, 120.9507°W), Twitchell Island West Pond (US-Tw1; 38.1074°N, 121.6469°W), Twitchell Island East End (US-Tw4; 38.1030°N, 121.6414°W), and Twitchell Island East Pond (US-Twt; 38.1072°N, 121.6426°W). CO$_2$ flux measurements and comparisons for other sites listed in Table 1 are shown in Supplemental Figs. S3 and S4.

Figure 5 compares the TROPOMI SIF retrievals to observations from AmeriFlux sites across California (see Table 1 and
Fig. 1 for the locations). The gray dots in Fig. 5 are all of the AmeriFlux GPP estimates and the black dots are those between
13:00-14:00 PST, similar to the TROPOMI overpass time (equatorial overpass time is 13:30 local solar time at nadir). This
overpass time is fortuitous in that it generally coincides with the daily maximum in GPP at the AmeriFlux sites. The green
dots are the actual TROPOMI SIF retrievals at that location that have the scene-specific relative weighting from the MODIS





NIR$_\text{v}$ (Eq. 5). No temporal smoothing has been applied in Fig. 5. We find a strong correspondence between SIF and GPP
across four different ecosystems. The top left panel shows that SIF clearly captures the onset of photosynthesis as well as the
punctuated seasonal cycle of GPP in a corn field (US-Bi2) with $r^2 = 0.79$. We also observe the gradual increase in GPP and
abrupt decline at a woody savanna site (US-Ton) and grassland site (US-Var) with $r^2 = 0.40$ and $r^2 = 0.59$, respectively. The
relatively high variability in SIF at US-Ton and US-Var from July to December (1-$\sigma$ spread of 0.33 mW/m$^2$/sr/nm) contrasts
the low variability during the dormant period at US-Bi2 and is likely associated with bright surfaces (implying a higher retrieval
uncertainty), quantifying the upper range of anticipatable noise. The bottom row of Fig. 5 shows a comparison of TROPOMI
SIF with GPP from three different wetland sites on Twitchell Island in the Sacramento-San Joaquin River Delta, we generally
find a strong correspondence between TROPOMI SIF and the three wetland sites ($r^2 = 0.42$, 0.48, and 0.29 for US-Tw1,
US-Tw4, and US-Tw5). The inter-site differences in GPP within a single ecosystem are larger than the SIF-GPP differences,
indicating some fine-scale heterogeneity that is likely not being captured here. In any case, the reasonable agreement with the
GPP at the wetland sites is encouraging because standing water can often bias reflectance-based indices, particularly in the
NIR (Gamon et al., 2013).
From this SIF-GPP comparison in Fig. 5, we infer a SIF-GPP scaling factor of 18.5±4.9 [($\mu$mol/m$^2$/s)/(mW/m$^2$/sr/nm)]
across the six sites in Fig. 5 (range of scaling factors is 13-25, see Fig. S3). Our comparison of TROPOMI SIF with GPP from
AmeriFlux sites in California indicates larger inter-ecosystem differences in the SIF-GPP relationship than intra-ecosystem
differences, lending credence to this universal scaling factor. However, there are two important caveats: 1) we do not have an
eddy covariance site in an evergreen forest, which is a major limitation as much of California is dominated by evergreen forests
and 2) we are not directly measuring GPP with SIF. As such, we refer to this SIF-estimated GPP as:
$\text{GPP}^* := 18.5 \cdot \text{SIF}.$ (9)
This single scaling from Eq. 9 seems to be a reasonable relation given the available information, with the caveat that there
could be differences between ecosystems that are unaccounted for.

## 5  Timing and spatial patterns of photosynthesis in California

Figure 6a shows the SIF-derived seasonal cycle of photosynthesis in California. One of the most prominent features is the
apparent double peak in the seasonal cycle. This double peak is present in both 2018 and 2019 with similar timing of the
maxima. The first peak occurs in April and the second peak occurs in June. Interestingly, the trough between the these peaks
occurs near the annual maximum in PAR (red line in Fig. 6d). This begs the question: *"What is driving this double peak in the*
*seasonality of California's photosynthesis?"*
We can use the CropScape database (see Fig. 1) to determine the ecosystems driving the spatio-temporal patterns in the
TROPOMI SIF data as it provides land cover classifications across the state of California at 30-m spatial resolution. However,
a notable limitation of the classifications from the CropScape data is the lack of discrimination for non-cropland areas. For
example, grasslands and pastures are combined into a single land type that seems to also include regions that would typically



**Figure 6. Seasonal cycle of photosynthesis in California. Panel A** shows the statewide mean SIF (black line) at 13:30 PST from November 2017 through September 2019 broken down by the contributions coming from cropland (yellow), evergreen forests (green), grasslands or pastures (purple), and other (gray). Rice is included in cropland here. Land types are taken from the 2018 CropScape database shown in Figure 1. Right axis shows the estimated GPP* based on comparison with AmeriFlux sites in California. **Panel B** shows the percentage of SIF coming from cropland (yellow). Vertical bars indicate the time periods with corresponding spatial plots in Panels E–G'. **Panel C** shows the vegetation indices (NDVI, EVI, and $NIR_v$) from MODIS over the same time period. **Panel D** shows clear sky PAR over California at 13:00 PST (dashed red line), surface PAR estimated from the ERA-Interim Reanalysis (thin red line), and cumulative precipitation over the water year from the GPM satellite (blue). **Panels E–G'** show the spatial patterns of SIF for the time periods indicated in Panel B.





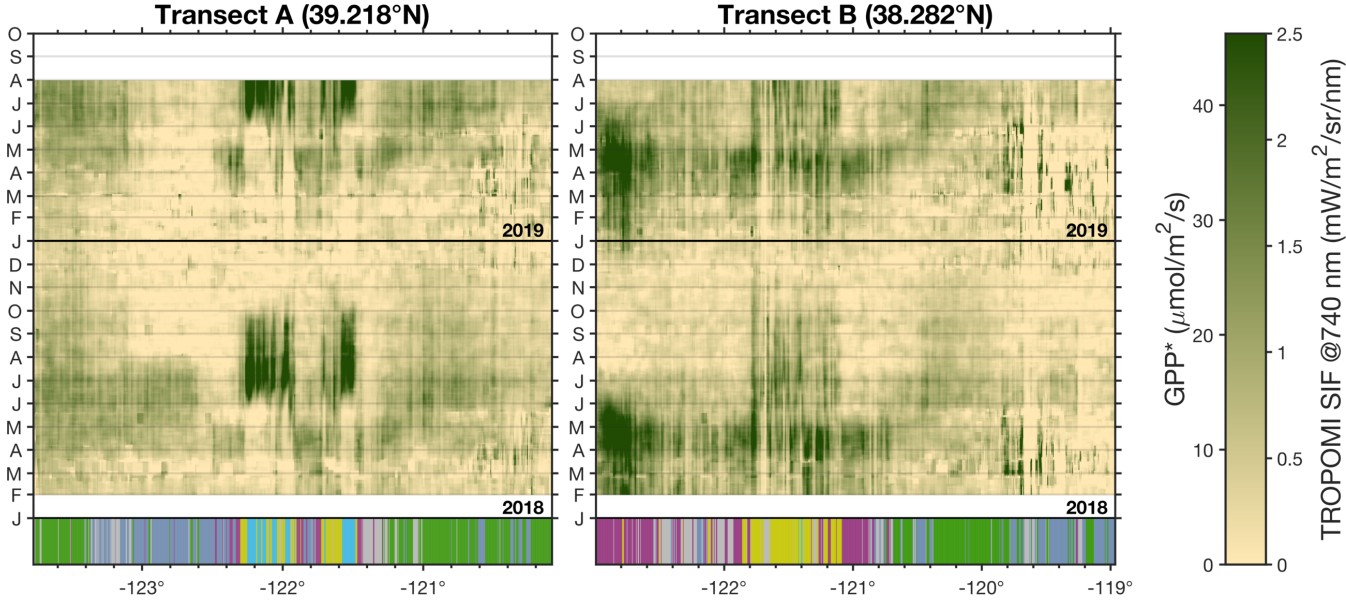

**Figure 7. Hovmöller diagrams for three transects across California.** Panels show Hovmöller diagrams from November 2017 through March 2019 for the two transects shown in Fig. 1. Bottom bar indicates the dominant land type with coloring from Fig. 1: green is evergreen forest, purple is grassland/pasture, cyan is rice, yellow is cropland (excluding rice), blue is shrubland, and gray is other.

be defined as oak savanna and chaparral. In lieu of a better sub-kilometer land cover dataset, we use the classifications from
the CropScape database for this work.
Figure 6a shows a breakdown of the regions contributing to the statewide SIF signal based on the land cover data from
the CropScape database. We find the California grasslands and pastures (a single classification that also includes chaparral
and oak savanna) have a single peak that coincides with the first statewide peak in April, this is consistent with the seasonal
cycle at California grassland sites in the AmeriFlux network (Fig. 5) that show a unimodal peak in the spring that ends in
May. Figs. 6e and 6e' show the mean spatial pattern in April 2018 and 2019, respectively, where we see that the April peak
coincides with a statewide increase in SIF. There are a few pertinent hotspots in grasslands or pastures during this April peak.
Notably, California's Central Valley and surrounding hills exhibit a strong photosynthesis signal in April. The valley to the east
of Bodega Bay (38.3°N, 122.9°W) appears as a large hotspot in both 2018 and 2019. This region lies on Transect B in Fig. 1
and the seasonal cycle is shown in more detail in Fig. 7.
The second peak in June shows a dominant contribution from evergreen forests (Fig. 6a). This can also be seen in the
spatial patterns from Fig. 6f and 6f' where the evergreen forests in Northern California exhibit a strong SIF signal. California's
Central Valley can be clearly distinguished as the surrounding hills have dried out (predominantly oak savanna and chaparral).
The observed photosynthesis from the Central Valley is maintained by heavily irrigated cropland throughout the valley.
The yellow line in Fig. 6b indicates the fraction of SIF in California that comes from cropland. We see the largest relative
contribution occurring in the fall. However, this is is primarily because all other ecosystems have gone dormant (see Fig. 6g)



as opposed to an increase in photosynthetic activity from cropland. The only region that shows an increase in photosynthesis
are the rice fields in the Sacramento Valley (the valley surrounding Sutter Buttes at 39.1°N, 121.5°W) in Northern California.
The rice fields show a SIF signal in excess of 2.5 mW/m$^2$/sr/nm during the fall (GPP* in excess of 45 $\mu$mol/m$^2$/s).

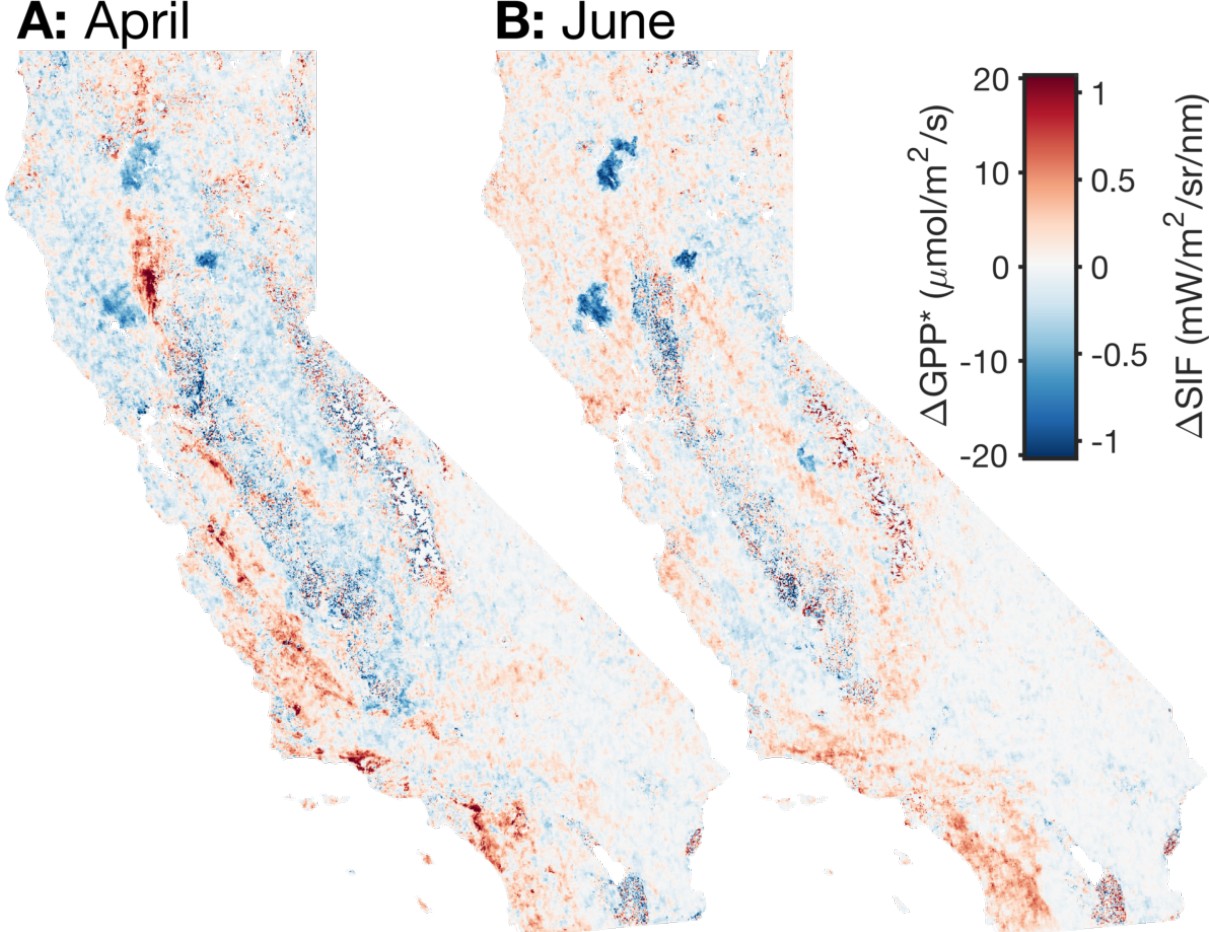

**Figure 8. Difference between 2019 and 2018. Panel A** shows the difference between the mean SIF in April 2019 and 2018. **Panel B** and **Panel C** are the same as panel A but for June and Fall (July 20-August 31), respectively. Red indicates higher SIF in 2019, blue indicates higher SIF in 2018.

Both 2018 and 2019 show a double peak in the seasonal cycle, however, the onset of the grassland driven peak differs
substantially between the two years. This difference is likely driven by the increased precipitation in 2019 (blue line in Fig. 6d).
There was 50% more precipitation in 2019 compared to 2018 and the precipitation occurred earlier in the water year. By mid-
February 2019 there was more precipitation than the annual total from 2018. This early precipitation allowed for an earlier and
longer growing season for the grasses. Figure 8 shows the difference in spatial patterns between 2019 and 2018. In general, we
find reasonable consistency between the two years. Most of the major differences are due to ecosystem disturbances such as



fires. The 2018 Northern California fires are a striking example of that (three large negative anomalies in Fig. 8b), the impact
of these fires is currently the focus of forthcoming work. An additional feature that stands out is the positive SIF anomaly in
Southern California, this increase in 2019 is due to the low rainfall the previous year.

4        Interestingly, none of the MODIS vegetation indices (NDVI, EVI, or NIR$_v$) show this double peak in photosynthesis

(Fig. 6c). The seasonal cycles from the three vegetation indices show a greening that starts in mid-winter (begins in De-
cember 2018) and increases roughly linearly to a peak in April. All three vegetation indices maintain that peak until July when
they show a roughly linear decline through the Fall. The seasonal cycle of the three MODIS vegetation indices bear a strong
similarity to the clear sky PAR seasonal cycle. This difference between SIF and the MODIS vegetation indices may be due to a
clear sky bias as the reflectance-based vegetation indices (NDVI, EVI, and NIR$_v$) can only be made under clear sky conditions,
whereas SIF can be retrieved in the presence of some clouds and aerosols (Frankenberg et al., 2011b). This is inferred by the
decline in PAR during May 2018 and May 2019 (Fig. 6d) that corresponds with a decline in SIF. This highlights one of the
differences between SIF and the MODIS vegetation indices, the vegetation indices are reflectance-based products whereas SIF
is a fluorescence signal emitted during photosynthesis and is thus coupled to the radiation regime. This again gets back to the
idea that SIF is measuring photosynthetic activity whereas the MODIS indices are measuring photosynthetic capacity.

15       Several ecophysiological reasons could also explain the SIF detection of a double peak feature, whereas MODIS vegetation

indices do not. Nearly 11% of the state of California consists of the California oak savanna (many in the foothills of coastal
mountains and the Sierras; Tyler et al., 2006). Over the course of the season, these ecosystems operate as an evergreen ecosys-
tem, whereby understory grass is photosynthetically active during the winter months, while trees (primarily oak species) reach
extremely high values of maximum carboxylation capacity ($V_{cmax}$) during the spring when water is plentiful, and then retain
their leaves throughout the summer in a highly photoprotective state (i.e., US-Ton; Xu and Baldocchi, 2003). Spatially, we
observe increased SIF values in oak savanna as well as chaparral ecosystems (also present on coastal and Sierra foothills) in
the early spring when available soil moisture is at a maximum (Xu and Baldocchi, 2004; Xu et al., 2004). As these ecosystems
enter the hot, dry summers, increases in sustained non-photochemical quenching and decreases in photochemistry result in
decreased fluorescence, while still appearing "green" to MODIS vegetation indices. Meanwhile, snow is melting rapidly at
higher elevations, making water available for many of the needleleaf evergreen species in the Sierras and Coastal ranges, then
the water resources become depleted and temperatures cool prompting these evergreen species to go back into a photoprotec-
tive state, resulting in a short photosynthetically active growing season that has been shown to be more well characterized by
SIF from GOME-2 than MODIS NDVI and EVI (Zuromski et al., 2018).

29       Figure 7 shows a Hovmöller diagram for three transects across Northern California (see Figure 1 for the location of the

transects). Transect A in Figure 7 shows the short but strong SIF signal from the rice fields. The timing of the SIF signal
from the rice fields agrees with the growing cycle for rice in California. Rice in the Sacramento Valley is typically planted in
mid-to-late May, the fields are then flooded, and harvested in mid-to-late September (University of California at Davis, 2018).
This observation of the rice fields is encouraging because we are observing photosynthesis in the presence of standing water,
which can often bias reflectance-based indices in the NIR (Gamon et al., 2013). In both 2018 and 2019 we observe the onset
of photosynthesis at the rice fields in the first few days of June and a rapid decline at the end of September. Transect B begins



in the valley to the east of Bodega Bay (location of the grassland hotspot) and crosses the central valley. This grassland hotspot
is present from April through May of both 2018 and 2019. The valley near Bodega Bay is dominated by pastures, however it
is currently unclear why this particular region exhibits a stronger SIF signal than other pastures in California. The persistent
strong signal in 2018 and 2019 might make it an interesting site for study with an eddy covariance site in the future.
## 6  Dominant "modes" of variability in California's photosynthesis

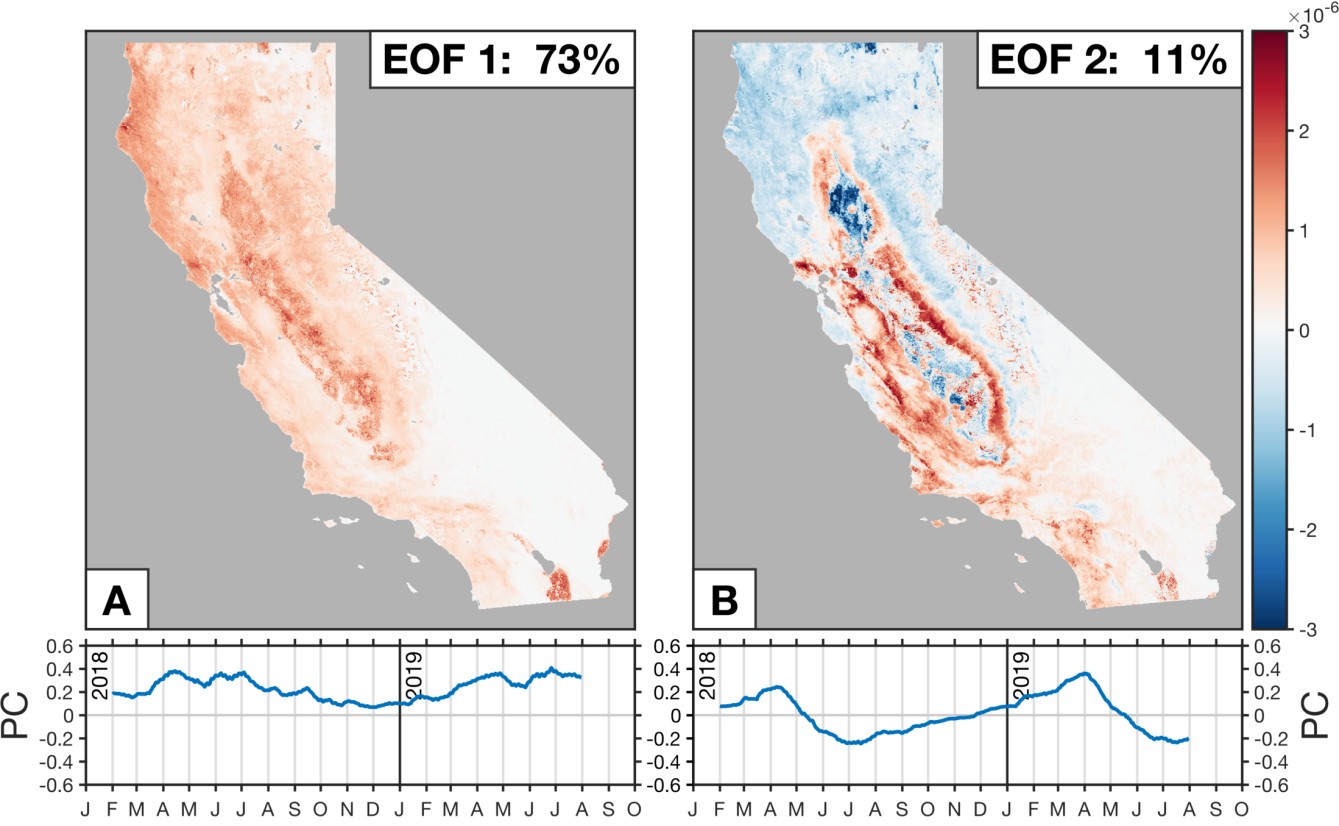

**Figure 9. EOFs and PCs for TROPOMI SIF over California.** Panels show the first two EOFs and PCs. EOFs are unit length (sum of the squares is equal to 1) and computed using unnormalized SIF spatially downscaled with $NIR_v$. Time series show the corresponding PC (blue line) from February 2018 through September 2019. Length of the PC is equal to the corresponding eigenvalue and has units of $mW/m^2/sr/nm$. Text in figure lists the percent of variance explained by that EOF. Fig. S5 shows the corresponding eigenvalue spectrum for the TROPOMI SIF data over California.

Section 5 discussed the spatio-temporal patterns for different regions and ecosystems, here we present an alternative method
of characterizing the dominant modes of spatio-temporal variability in photosynthesis using Empirical Orthogonal Functions
(EOFs) and their associated Principal Components (PCs). EOFs are a matrix factorization that are commonly used to identify





structure in a spatial dataset and yield a finite number of modes. These modes compactly represent the data and are often
interpreted as physical modes of the system.
Figure 9 shows the first two EOFs and their associated PCs for the TROPOMI SIF data over California, the corresponding
eigenvalue spectrum can be seen in Fig. S5. The first two EOFs corroborate the findings from Section 5 and, taken together,
explain 84% of the variability in the TROPOMI SIF data:
– EOF 1: the mean signal
– EOF 2: the double peak
The first EOF (Fig. 9a) represents the mean signal and explains 73% of the variability in the TROPOMI SIF data. From the
spatial pattern we can see that it includes most of the biomass in California and is strongly correlated to the state-wide mean
SIF: $r^2 = 0.99$. The associated principal component bears a strong similarity to the statewide mean SIF seasonal cycle (Fig. 6a).
This finding is not entirely surprising because we are using un-normalized SIF data for the matrix factorization. This means
that the most important mode of variability is the mean signal and that the following EOFs are anomalies relative to the mean
signal.
The second EOF (Fig. 9b) represents the double peak in the timing of California's photosynthesis. This EOF combines the
signal from the grasslands (positive phase of EOF 2) and the evergreen forests (negative phase of EOF 2). We find EOF 2
to be positively correlated with the grassland fraction from the CropScape database ($r = 0.55$) and negatively correlated with
the evergreen forests ($r = -0.37$). There is also a negative correlation with the rice fields ($r = -0.31$). The associated principle
component serves to amplify the seasonal cycle from EOF 1 in grasslands during April and amplify the forest peak in June.
This is because the red region (grasslands) in Fig. 9b will contribute a positive anomaly in April and a negative anomaly in June.
Conversely, the blue region (evergreen forests and rice) will contribute a negative anomaly in April and a positive anomaly in
June. This EOF arises because the grasslands and forests are both spatially separated and out of phase with each other, allowing
the matrix factorization to place them into a single EOF that represents the processes driving the double peak in the timing of
California's photosynthesis.
It should be noted that these EOF patterns found here are unlikely to be true "physical modes" (see, for example, Monahan
et al., 2009). That is to say, we would not necessarily expect the response to a perturbation to follow patterns shown in Fig. 9.
EOF 2 is a good example of this because it seems unlikely that the grasslands and forests will exhibit opposing responses to a
forcing. Grasslands and forests are combined into a single EOF simply because there is little loss of information by combining
them due to the spatial separation and phase offset. This is not to argue against the utility of EOFs. EOFs are a useful method
for identifying structure in geophysical datasets, as evidenced here by their identification of the double peak in the timing of
California's photosynthesis.





## 7 Conclusions

We present an oversampling and downscaling method to obtain daily estimates of Solar Induced chlorophyll Fluorescence (SIF), a proxy for photosynthetic activity, at 500-m spatial resolution from TROPOMI. To our knowledge, this is the highest spatial resolution SIF dataset from satellite measurements. We find a double peak in the seasonality of photosynthesis in California during 2018 and 2019, a feature that is not present in the MODIS vegetation indices (NDVI, EVI, or $NIR_v$). Analysis of the spatial and temporal patterns of the SIF data indicates that the double peak is due to two processes that are out of phase with each other: woody grasslands (e.g., grasslands, chaparral, and oak savanna) and evergreen forests.

Our work applies methods developed for previous satellite retrievals (oversampling) and uses estimates of sub-grid scale vegetation (downscaling) to obtain daily 500-m spatial resolution SIF from TROPOMI over California. The oversampling method results in a smooth spatial field and removes artifacts due to complex topography and the wide TROPOMI swath. The downscaling method further refines the high resolution spatial patterns by bringing in *a priori* information on the sub-grid vegetation patterns. The oversampling and downscaling methods do not alter the large scale spatio-temporal patterns as they conserve the SIF signal over a single scene.

TROPOMI SIF data and MODIS vegetation indices are reasonably consistent at annual timescales over California, but show weaker relationships at daily and monthly timescales. This implies that TROPOMI SIF is providing some information that is distinct from the MODIS vegetation indices. TROPOMI SIF data show a strong correspondence with half-hourly estimates of GPP at multiple AmeriFlux sites across different ecosystems including: cropland, grassland, savanna, and wetlands. We find a linear relationship between SIF and GPP that is largely invariant across ecosystems with an intercept that is not significantly different from zero. As such, we use SIF as an estimate of GPP* with the caveat that some ecosystems are not represented in our California analysis.

TROPOMI SIF data show a double peak in the seasonality of photosynthesis in California, a feature that is not present in the MODIS vegetation indices. The double peak in the seasonality of California's photosynthesis is due to two processes that are out of phase with each other: grasses show a maximum in April and evergreen forests peak in June. An empirical orthogonal function (EOF) analysis corroborates the phase offset and spatial patterns driving the double peak. The EOF analysis also indicates that two spatio-temporal patterns explain 84% of the variability in the TROPOMI SIF data.

The results shown here are promising for obtaining global near-daily GPP at sub-kilometer spatial scales using satellite measurements. This, in turn, may prove helpful in addressing long-standing questions regarding the mechanisms and locations driving carbon uptake in the Northern Hemisphere. It would also allow us to monitor climate change impacts on vulnerable ecosystems at local-to-global scales.

*Acknowledgements.* A. J. Turner is supported as a Miller Fellow with the Miller Institute for Basic Research in Science at UC Berkeley. R. C. Cohen acknowledges support from the TEMPO project SV3-83019. P. Köhler and C. Frankenberg are funded by the Earth Science U.S. participating investigator (grant: NNX15AH95G). This research used the Savio computational cluster resource provided by the Berkeley Research Computing program at the University of California, Berkeley (supported by the UC Berkeley Chancellor, Vice Chancellor for



Research, and Chief Information Officer). This research also used resources from the National Energy Research Scientific Computing Center,
which is supported by the Office of Science of the U.S. Department of Energy under Contract No. DE-AC02-05CH11231. TROPOMI SIF and
MODIS NBAR data are publicly available at "ftp://fluo.gps.caltech.edu/data/tropomi/" and "https://e4ftl01.cr.usgs.gov/MOTA/MCD43A4.006/",
respectively. Funding for AmeriFlux data resources was provided by the U.S. Department of Energy's Office of Science. We would like to
thank Dennis Baldocchi (UC Berkeley) for sharing the AmeriFlux data and for providing extensive feedback on the work. Finally, we are
extremely grateful to the team that has realized the TROPOMI instrument, consisting of the partnership between Airbus Defence and Space,
KNMI, SRON, and TNO, commissioned by NSO and ESA.




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
