# Peer review of "A double peak in the seasonality of California's photosynthesis as observed from space"

_Biogeosciences, 2019_

## Referee Comment (RC1) · Luis Guanter (Referee) · 21 Oct 2019

The manuscript by Turner et al. presents a study of the potential of sun-induced fluorescence (SIF) derived from the S5P/TROPOMI mission to track spatial and temporal variations of photosynthesis within California's ecosystems. The authors first downscale the original TROPOMI SIF retrievals from ∼3.5x7 km2 to 500 m and then compare the resulting signal with gross primary production (GPP) estimates derived at flux towers as well as with MODIS vegetation indices for the entire California area. They find that the seasonality of SIF at the state level shows a double peak which is not found in equivalent time series from MODIS indices.

In my opinion this is an interesting study with some potentially important messages.

[Figure]

First, it presents a simple methodology to downscale the new SIF data from TROPOMI to MODIS like spatial sampling, which I can imagine it will be reproduced by many others in the future; second, it takes advantage of this novel data set to show that the spatio-temporal variability of photosynthesis in California is driven by that of two groups of biomes for which photosynthesis peaks in different times of the year. This finding is used to illustrate the superior sensitivity of SIF to actual photosynthetic processes than reflectance-based indices, which are good proxies of photosynthetic potential.

The manuscript is very well written and presented, methods and data are innovative and the results are interesting, so I recommend it for publication in Biogeosciences. Before that, however, I would appreciate if the authors could address the following two points in their revision of the manuscript:

1.Double peak, PAR and/or physiology? The authors acknowledge that the different seasonality in SIF and vegetation indices may be due to a clear-sky bias in the vegetation indices, but also claim that "SIF can detect the downregulation of photosynthesis even when plants appear green", which seems to hint that it is not only the reaction to solar irradiance which makes SIF to show the double-peaked seasonality. To substantiate this claim, it would be interesting to see a plot of NIRv x PAR_ground (with PAR_ground the at-surface PAR for all-sky conditions), and evaluate to what extent its seasonality resembles that of SIF. The difference between SIF and NIRvxPAR could be attributed to physiological effects captured by SIF.

2.GPP scaling - The authors scale SIF to GPP as GPP*=18.5xSIF. However, I think we know better. There have been a number of papers in the last years showing that factors such as a canopy structure, photosynthetic pathway or observation geometry affect the SIF-GPP relationship making the use of a global scaling factor to be questionable. On the other hand, this study is based on the analysis of time series and no quantification of GPP is performed, so I don't see why the authors need to scale SIF to GPP values. I would therefore recommend the authors to simply use SIF rather than both SIF and GPP* in the analysis (Figs. 6 to 8).

Other comments:

- p1, L3 "oversampling and downscaling" –> simply "downscaling" would probably be more clear for most of the readers.

- p2, L3 "more direct methods" than what methods?

- p2, L24 Several SIF downscaling methods have been published in the last years which are actually not based on machine learning nor intended to produce spatially-continuous SIF data sets from OCO-2 SIF retrievals. In particular, the method by Duveiller et al. to downscale GOME-2 SIF to 0.05° (last implementation here https://www.earth-syst-sci-data-discuss.net/essd-2019-121/) could also be adapted to TROPOMI. Please, discuss pros and cons of the oversampling/downscaling method presented in this manuscript with respect to that by Duveiller et al. and any other comparable downscaling method.

- p4, L10 "near-infrared and shortwave infrared".

- p4, L19 "The TROPOMI SIF retrieval uses..." I don't think the average reader will understand this sentence without any further introduction to PCA-based SIF retrievals.

- p4, L 25 I can't find any information on cloud filtering, so I assume that the authors are simply not applying any. Please, discuss this here, e.g. whether no cloud filtering could/should applied globally when using SIF as a proxy for GPP (which would be somewhat scary...).

- Fig 8. Panel C?

---

## Short Comment (SC1) · 2 Nov 2019

This review was prepared as part of graduate program course work at Wageningen University, and has been produced under supervision of Prof Wouter Peters. The review has been posted because of its good quality, and likely usefulness to the authors and editor. This review was not solicited by the journal.

Please also note the supplement to this comment:
https://www.biogeosciences-discuss.net/bg-2019-387/bg-2019-387-SC1-supplement.pdf

**Supplement:**

This manuscript presents the applicability of remotely sensed SIF datasets to explain variability and seasonal dynamics of photosynthesis. Oversampling and downscaling manipulations on the TROPOMI SIF dataset were used to acquire daily estimates of SIF. Two major advantages of SIF over MODIS datasets are its low sensitivity to clouds or aerosols and its ability to detect a decrease in photosynthesis even when plants stay green. A double peak in the seasonal variability of SIF was found and linked with out-of-phase vegetation photosynthesis: woody grasslands and evergreen forests. This double peak was not found in MODIS datasets, even though there was strong correlation between MODIS and SIF on the annual timescale. This correlation was significantly weaker at daily and monthly timescales. Variability in spatio-temporal patterns and its relation with variability in TROPOMI SIF data was determined by using EOFs. These EOFs confirmed the double peak and showed that spatio-temporal vegetation patterns explain 84% of the variability in SIF data.

In my opinion, the study is interesting and introduces a relevant novelty in the narrow scientific community bridging remote sensing science and photosynthesis research. However, I think three flaws are present in the current manuscript, of which I recommend some revisions before publication.

To start with the first issue, on page 9 (lines 7-27) a fourteen-day moving window is used in combination with a spatial downscaling method to obtain daily estimates of SIF at a high resolution. In combination with the consequent pre-weighting of the SIF signal by the underlying vegetation fraction (MODIS NIRv), large-scale changes in spatio-temporal patterns are conserved. On lines 20-21, the authors assume that the observed SIF from TROPOMI likely originates from more vegetated regions within that scene. However, the $R^2$ value of the linear correlation between SIF and NIRv (0.52, Figure 2 on page 7) implies that a significant part of the variance in SIF cannot be explained by the underlying vegetation fraction. Besides, by using the averaged value of the 14-day moving window, a pseudo-daily average SIF value is created, rather than the actual daily value. This is fine, provided that a certain accuracy assessment is conducted. Especially because the authors mention, on page 9 lines 31-32, significant differences are found with the similar method of Köhler et al. (2018) in which a quality control and accuracy assessment are indeed present. In addition, the downscaling (from 24.5 $km^2$ to 0.25 $km^2$) is likely to introduce inaccuracies, which requires quantification.

Therefore, it is recommended that the authors provide more evidence that observed differences in SIF are indeed linked to more vegetated regions, rather than barren areas, water bodies or other external factors. Liu et al. (2019) should be consulted, who state that scattering and re-absorption of sunlight into the leaves means that the SIF measured at canopy level is only part of the total ecosystem SIF emission.

Secondly, from page 11 onwards, the authors use a method to infer GPP from SIF, based on light-use efficiency and the probability of SIF photons escaping the canopy. Interestingly, Paul-Limognes et al. (2018) found that SIF was more affected by environmental conditions than GPP. Midday-depressions in SIF were linked to peak VPD values following peak photosynthetic photon flux density (PPFD). Besides, Walther et al. (2016) state that in evergreen needle-leaf forests, the length of the photosynthetically active period indicated by SIF is up to six weeks longer and commences a month earlier than the green biomass changing period proxied by EVI. Even though the authors used $NIR_v$ instead of EVI to downscale SIF, the different timing could significantly alter the double peak structure. Moreover, the authors state there is a lack of GPP measurements in evergreen forests, while much of California is dominated by this vegetation type (page 13, line 17-19). In combination with the a-synchronous SIF/MODIS dynamics, this will propagate into a major bias in the scaling factor of 18.5 ± 4.9 which is inferred on page 13, line 14. Therefore, I think that the equation on page 13, line 20 ($GPP := 18.5 * SIF$) should include a revised quantification of the error margins. In doing so, the authors

should determine an alternative error margin whilst taking into account the fractional contribution of evergreen forests to GPP. The latter can best be inferred from a biosphere model or studies which used eddy-covariance measures in similar evergreen forests.

Lastly, the authors successfully identify a double-peak in the seasonality of GPP. However, the number of (recent) references concerning underlying reasons for this double peak or other case studies in which a double peak is found, is unsatisfactory. References to Xu and Baldocchi (2003), Xu et al. (2004), Xu and Baldocci (2004) explain changes in carbon fluxes between ecosystems and vegetation types well, yet the link with SIF dynamics is lacking (Page 17, lines 15-22). Perhaps the following is a cause of the state-of-the-art novelty of this subject, but there are zero references made to any other recent papers discovering the double peak in GPP/SIF. Given the importance of this conclusion to the subject of the manuscript, I highly suggest investigating and mentioning recent existing literature explaining the double peak phenomenon. If the latter turns out to be infeasible because it is such a novelty, it is suggested to emphasise the scientific novelty in this paper. For instance, Li et al. (2014) imply that MODIS EVI is unsuitable for detecting a double peak in vegetated areas which usually manifest double peaks. This would strengthen the relevance as to why SIF needs to be used.

**Minor arguments and small revisions**

Minor issue 1: In Table 1, all vegetation types have two or more study sites except for the WSA (Woody Savannas). I would like to give the authors awareness that one study site might not be representative for the entire ecosystems, especially when all other vegetation types have multiple sites.

In Figure 1, Page 3: The description mentioned that black stars show the location of six Ameriflux sites, However I can only discern three and they seem to be closely packed at this resolution.

In Figure 2 on page 7, the axes lack titles. This is relevant to include for the x-axes of the bottom row of graphs, as the range of the axes are different.

In Figure 3 on page 8 the swath resolution is 4.0 km x 7.0 km, whereas in the text on Page 9, line 4 it is stated that this resolution is 3.5 km x 7.0 km. This should match.

On page 9, line 19-20: perhaps it is necessary to introduce that the $NIR_v$ was used in the pre-weighting of SIF, rather than introducing it later on Page 11, line 5.

On page 14 in the figure description, a reference to Panel G' is made, whereas this panel is not present in the accompanying figure (6).

On page 16, line 8-9 it is stated that a 'reasonable consistency' is found. This should be quantified.

In the conclusion on page 20, parts of line 6-7 and line 22-23 have very similar information.

**References**

Köhler, P., Frankenberg, C., Magney, T. S., Guanter, L., Joiner, J., & Landgraf, J. (2018). Global retrievals of solar-induced chlorophyll fluorescence with TROPOMI: First results and intersensor comparison to OCO-2. Geophysical Research Letters, 45(19), 10-456.

Li, L., Friedl, M., Xin, Q., Gray, J., Pan, Y., & Frolking, S. (2014). Mapping crop cycles in China using MODIS-EVI time series. Remote Sensing, 6(3), 2473-2493.

Liu, X., Guanter, L., Liu, L., Damm, A., Malenovský, Z., Rascher, U., ... & Gastellu-Etchegorry, J. P. (2019). Downscaling of solar-induced chlorophyll fluorescence from canopy level to photosystem level using a random forest model. Remote Sensing of Environment, 231, 110772.

Paul-Limoges, E., Damm, A., Hueni, A., Liebisch, F., Eugster, W., Schaepman, M. E., & Buchmann, N. (2018). Effect of environmental conditions on sun-induced fluorescence in a mixed forest and a cropland. Remote sensing of environment, 219, 310-323.

Walther, S., Voigt, M., Thum, T., Gonsamo, A., Zhang, Y., Köhler, P., ... & Guanter, L. (2016). Satellite chlorophyll fluorescence measurements reveal large-scale decoupling of photosynthesis and greenness dynamics in boreal evergreen forests. Global change biology, 22(9), 2979-2996.

---

## Referee Comment (RC2) · Anonymous Referee #2 · 7 Nov 2019

The authors of this manuscript present work in which they use recent SIF retrievals from the Tropomi sensor over California and techniques of oversampling and downscaling to obtain a fine spatial resolution of 500m. They compare those to flux tower-derived GPP estimates and find generally good agreement and a linear relationship, which they use to scale SIF to obtain a spatial GPP estimate from SIF. Averaged over the whole state of California they present a double-peak in the seasonal cycle of SIF that is present in both 2018 and 2019 and can be explained by different phasing of vegetation activity in different vegetation types (as shown by time series per biome and by decomposition via EOF analysis). The trajectory of the greenness indices is very dissimilar and the authors explain this with a possible clear sky bias and the fact that SIF is more closely linked to photosynthesis in contrast to vegetation indices.

[Figure]

This work is highly interesting and its the main contribution to the scientific community is twofold: a) they introduce novel methods of oversampling and downscaling to the SIF community which offers the exciting opportunity to analyse SIF data at unprecedented spatial resolution. They also raise awareness of possible prominent retrieval biases related to bright surfaces. b) The different dynamics between canopy greenness and photosynthetic activity and the resulting benefits of SIF in general, and of Tropomi SIF in particular, to track GPP dynamics.

However, I have two major concerns regarding the second point and main message of the paper which in my opinion are necessary to address before a publication in Biogeosciences:

First, it is not fully clear from the work description if the comparison between greenness and SIF is meaningful: Have any filters been applied to either tower measurements, SIF or the MODIS data to remove low quality data? Also, the authors mention a clear-sky bias of the reflectance measurements as a possible explanation for the different dynamics. This implies to me that the data of VIs and SIF have not been matched in space and time before aggregating to the spacial averages shown in Fig.6. If this was indeed the case, the time series are representative of different places and therefore not fully comparable. I would like to ask the authors to clarify and if need be, to improve on this point to corroborate the main message of the paper.

Second, I see the explanation for the different dynamics in SIF and greenness as incomplete. The authors convincingly argue that the different phasing of activity between evergreen forests and grasses, chaparral, and oak savanna causes the double peak in SIF. However, a similar decomposition by land cover type (Fig.6a) is missing for the greenness indices and I strongly suggest to include this in the analysis (at least for one of the indices) in order to get an idea of where/ in which ecosystems SIF and greenness are particularly dissimilar (Fig.6c). Otherwise, a sentence like in the abstract that SIF 'can detect the downregulation of photosynthesis even when plants appear green' is not justified from the material presented in the paper. Finally, this analysis of where

and when VIs and SIF disagree, could be completed by a driver analysis to understand which processes does SIF see that greenness does not and to undermine the argumentation in p.17 ll.15-28. There are features as those in May in both years, which coincide with similar dips in light and rain events, it is not clear which of these is more important for which ecosystem. There are other prominent features such as the smaller peak in September 2018 in SIF which does not seem to have an obvious relationship with either precipitation or light.

Overall, the paper is very interesting, well structured, very clearly written with detailed explanations and very appealing figures. It can be streamlined though. The scaling from SIF to GPP based on the tower measurements is not necessary for this manuscript and a distracting side information, especially given the weaknesses of the scaling that the authors discuss. I suggest to remove this and list further minor comments below.

Minor comments:

- The higher correlation between greenness and SIF at longer time scales is mentioned both in the abstract and conclusions but in the main text only in a sub-clause, and is not a main finding of your work. I see this as distracting side information as well, which does not necessarily need to be mentioned in both the abstract and the conclusions.

-Abstract: 'The different seasonality in the vegetation indices may be due to a clear-sky bias in the vegetation indices, whereas SIF has a low sensitivity to clouds and can detect the downregulation of photosynthesis even when plants appear green.' This sentence illustrates my major comment from above that the question of what drives the SIF response in the different ecosystems is not sufficiently covered by the analysis.

-The fact that there is a double peak in SIF but not in the VIs is mentioned twice in the conclusions.

- Apart from the fact, that the scaling from SIF to GPP is not needed in this manuscript,

it is rather uncommon to use the unit of mu mol/m2/s from the tower measurements also for seasonal values as in the maps in Fig.6. gC/m2/day is rather common.

- Fig.6G' does not exist, pay attention in caption and Fig6b.

-p.9 l.34: Can you really resolve daily features with an average over 14 days despite daily sampling?

-p.11 l.29 -p.12 l.2: To my (admittedly non-native English) ears the word 'owing' in this sentence sounds misplaced.

- Fig.6B: why is the cropland contribution stressed in this panel?

-p.19 l.26: 'it seems unlikely that the grasslands and forests will exhibit opposing responses to a forcing.' It probably depends and an extended analysis as suggested above can give indications of whether this is true for California or not. There are counter examples e.g. in Flach et al. 2018 https://doi.org/10.5194/bg-15-6067-2018 or Walther et al. 2019 https://doi.org/10.1029/2018GL080535

---

## Author Comment (AC2) · 22 Nov 2019

We have responded to the reviewer comments in the attached supplement.

Please also note the supplement to this comment:
https://www.biogeosciences-discuss.net/bg-2019-387/bg-2019-387-AC2-
supplement.pdf
* * *

---

## Author Comment (AC3) · 22 Nov 2019

We have responded to the reviewer comments in the attached supplement.

Please also note the supplement to this comment:
https://www.biogeosciences-discuss.net/bg-2019-387/bg-2019-387-AC3-supplement.pdf

---

## Author Response (AR1)

**Response to Reviewer and Short Comments:**

We thank the Editor (Dr. Martin De Kauwe), Professor Luis Guanter, the Anonymous Reviewer, and Mr. Paolo Tasseron for their time and constructive comments on our manuscript.
* * *
**Editor Comments:**

*If I could make one suggestion, which likely reflects my personal world bias. I felt like more could be done to provide a mechanistic interpretation of the double peak. For example, why would photosynthetic capacity drop and then peak again? One interpretation is differences in PAR, although I see little evidence from your Fig 6? Another is more physiological, is it a response to increasing moisture stress? You could conceivably show this by examining timeseries of air temperature, VPD (if you have it from nearby) and perhaps a climatic water deficit (e.g. PPT-PET)? Or at least, show something about precipitation that might correspond with this double peak. Currently, you show an accumulated PPT, which I don't think shows a mechanistic explanation.*

Respectfully, we feel like we have provided a mechanistic interpretation of the double peak. Photosynthetic capacity has a unimodal response for the grassland+chaparral+oak savanna regions. Photosynthetic capacity also has a unimodal response for evergreen forests.
- **Peak 1 (April):** grassland + chaparral + oak savanna
- **Peak 2 (June):** evergreen forests

The processes driving these two peaks are distinct, but we then find a bimodal seasonal cycle when we look at the statewide mean because we see these two peaks. We then go on to discuss several ecophysiological reasons why this could manifest itself in a statewide double peak feature in Section 5. Finally, Section 6 (the EOF/PC analysis) further corroborates this finding by demonstrating that distinct regions in California drive the two peaks.

Page 1, Line 12: *"The double peak in the seasonality of California's photosynthesis is due to two processes that are out of phase: grasses, chaparral, and oak savanna ecosystems show an April maximum while evergreen forests peak in June."*

Page 17, Line 18: *"Several ecophysiological reasons could also explain the SIF detection of a double peak feature, whereas MODIS vegetation indices do not. Nearly 11% of the state of California consists of the California oak savanna (many in the foothills of coastal mountains and the Sierras; Tyler et al., 2006). Over the course of the season, these ecosystems operate as an evergreen ecosystem, whereby understory grass is photosynthetically active during the winter months, while trees (primarily oak species) reach extremely high values of maximum carboxylation capacity ($V_{cmax}$) during the spring when water is plentiful, and then retain their leaves throughout the summer in a highly photoprotective state (i.e., US-Ton; Xu and Baldocchi, 2003). Spatially, we observe increased SIF values in oak savanna as well as chaparral ecosystems (also present on coastal and Sierra foothills) in the early spring when available soil moisture is at a maximum*

*(Xu and Baldocchi, 2004a; Xu et al., 2004). As these ecosystems enter the hot, dry summers, increases in sustained non-photochemical quenching and decreases in photochemistry result in decreased fluorescence, while still appearing "green" to MODIS vegetation indices. Meanwhile, snow is melting rapidly at higher elevations, making water available for many of the needleleaf evergreen species in the Sierras and Coastal ranges, then the water resources become depleted and temperatures cool prompting these evergreen species to go back into a photoprotective state, resulting in a short photosynthetically active growing season that has been shown to be more well characterized by SIF from GOME-2 than MODIS NDVI and EVI (Zuromski et al., 2018)."*

Page 18, Line 27: *"The second EOF (Fig. 9b) represents the double peak in the timing of California's photosynthesis. This EOF combines the signal from the grasslands (positive phase of EOF 2) and the evergreen forests (negative phase of EOF 2). We find EOF 2 to be positively correlated with the grassland fraction from the CropScape database (r = 0.55) and negatively correlated with the evergreen forests (r = -0.36). There is also a negative correlation with the rice fields (r = -0.32). The associated principle component serves to amplify the seasonal cycle from EOF 1 in grasslands during April and amplify the forest peak in June. This is because the red region (grasslands) in Fig. 9b will contribute a positive anomaly in April and a negative anomaly in June. Conversely, the blue region (evergreen forests and rice) will contribute a negative anomaly in April and a positive anomaly in June."*

Regarding the NIRv x PAR point raised by Professor Guanter, I think that even if the authors are not going to show such a plot (for the reasons outlined), it would be very worthwhile adding some discussion text on this point.

See response to Comment #1 from Professor Guanter. We have also added the following text to the main text and a paragraph in the supplement.

Page 17, Line 31: *"Future work comparing SIF and MODIS indices with measured PAR at AmeriFlux sites would be useful in further evaluating the role of radiation and physiology in the double peak feature."*

Supplemental Section S1: *"The two peaks in the California photosynthesis seasonal cycle coincide with a slight decline in PAR inferred from ERA Interim, a time when the MODIS vegetation indices (e.g., $NIR_v$) remain nearly constant. If part of the difference between the SIF seasonal cycle and $NIR_v$ seasonal cycle is indeed due to a clear sky bias as we mention, then comparing SIF with $NIR_v \times PAR$ would be the more appropriate comparison. However, the all-sky PAR dataset used in our work (ERA Interim) has a known issue that makes it unreliable (see known issue number 2: "https://confluence.ecmwf.int/display/CKB/ERA-Interim+known+issues"). The clear sky PAR from ERA Interim is reliable and, as such, we have applied a correction to the statewide PAR based on the reliable clear sky PAR but we are hesitant to draw any conclusions using this scaling at finer scales. All this is to say, the PAR data presented in main text Figure 6 is illustrative of potential reductions in PAR during May when there is a difference in the seasonal cycles from SIF and $NIR_v$, but we are wary of using it to directly scale $NIR_v$ and/or compare with SIF. Further study of SIF and $NIR_v$ in other regions is obviously needed."*

**Reviewer #1 (Professor Luis Guanter) Comments:**

*The manuscript is very well written and presented, methods and data are innovative and the results are interesting, so I recommend it for publication in Biogeosciences. Before that, however, I would appreciate if the authors could address the following two points in their revision of the manuscript:*

We thank Professor Guanter for his feedback and comments on the work.

**General Comments**

**1.)** Double peak, PAR and/or physiology? The authors acknowledge that the different seasonality in SIF and vegetation indices may be due to a clear-sky bias in the vegetation indices, but also claim that *"SIF can detect the downregulation of photosynthesis even when plants appear green"*, which seems to hint that it is not only the reaction to solar irradiance which makes SIF to show the double-peaked seasonality. To substantiate this claim, it would be interesting to see a plot of $NIR_v \times PAR_{ground}$ (with $PAR_{ground}$ the at-surface PAR for all-sky conditions), and evaluate to what extent its seasonality resembles that of SIF. The difference between SIF and $NIR_v \times PAR$ could be attributed to physiological effects captured by SIF.

The line *"SIF can detect the downregulation of photosynthesis even when plants appear green"* was based on inferences from previous work, not inferred here. The abstract has been amended to highlight this.

Page 1, Line 9: *"The different seasonality in the vegetation indices may be due to a clear-sky bias in the vegetation indices, whereas previous work has shown SIF to have a low sensitivity to clouds and to detect the downregulation of photosynthesis even when plants appear green."*

We appreciate the suggestion from Professor Guanter to show $NIR_v \times PAR$, however there are some caveats with the available PAR data that make such a comparison unreliable. Specifically, there are known issues with the all-sky PAR data from ERA Interim: "https://confluence.ecmwf.int/display/CKB/ERA-Interim+known+issues" (see known issue number 2). The clear sky PAR from ERA Interim is reliable and we have applied a correction to the statewide PAR based on the reliable clear sky PAR but we are hesitant to draw any conclusions using this scaling at finer scales. In a similar vein, the Badgely *et al.*, GBC (2019) paper found that $NIR_v \times PAR$ worked well for predicting GPP at FLUXNET sites if they used measured PAR, but using global PAR datasets actually yielded worse estimates than if they did not include PAR as a predictor (personal communication with co-first author Lee Anderegg, UC Berkeley). This is because the global PAR datasets are poor. All this is to say, the PAR presented in Figure 6 is illustrative of potential reductions in PAR during May, but we are wary of using it to directly scale $NIR_v$ and/or compare with SIF. Further study of SIF and $NIR_v$ in other regions is obviously needed.

Page 17, Line 31: *"Future work comparing SIF and MODIS indices with measured PAR at AmeriFlux sites would be useful in further evaluating the role of radiation and physiology in the double peak feature."*

Supplemental Section S1: *"The two peaks in the California photosynthesis seasonal cycle coincide with a slight decline in PAR inferred from ERA Interim, a time when the MODIS vegetation indices (e.g., $NIR_v$) remain nearly constant.  If part of the difference between the SIF seasonal cycle and $NIR_v$ seasonal cycle is indeed due to a clear sky bias as we mention, then comparing SIF with $NIR_v \times PAR$ would be the more appropriate comparison.  However, the all-sky PAR dataset used in our work (ERA Interim) has a known issue that makes it unreliable  (see known issue number 2: "https://confluence.ecmwf.int/display/CKB/ERA-Interim+known+issues").  The clear sky PAR from ERA Interim is reliable and, as such, we have applied a correction to the statewide PAR based on the reliable clear sky PAR but we are hesitant to draw any conclusions using this scaling at finer scales.  All this is to say, the PAR data presented in main text Figure 6 is illustrative of potential reductions in PAR during May when there is a difference in the seasonal cycles from SIF and $NIR_v$, but we are wary of using it to directly scale $NIR_v$ and/or compare with SIF.  Further study of SIF and $NIR_v$ in other regions is obviously needed."*

**2.)**  GPP scaling - The authors scale SIF to GPP as GPP*=18.5×SIF. However, I think we know better. There have been a number of papers in the last years showing that factors such as a canopy structure, photosynthetic pathway or observation geometry affect the SIF-GPP relationship making the use of a global scaling factor to be questionable. On the other hand, this study is based on the analysis of time series and no quantification of GPP is performed, so I don't see why the authors need to scale SIF to GPP values. I would therefore recommend the authors to simply use SIF rather than both SIF and GPP* in the analysis (Figs. 6 to 8).

Our reasoning for showing GPP* is to remind the reader of the major motivation for the use of SIF: to study carbon uptake.  We acknowledge the shortcomings of our SIF-GPP relationship (i.e., the lack of eddy flux sites in important ecosystems) and put an asterisk on our GPP variable to emphasize that.  We feel that his is a fair representation of the caveats while also highlighting the ultimate aim of work using SIF.

Page 13, Line 23: *"To reiterate, there is a clear correspondence between the observed SIF and GPP estimated for the different AmeriFlux sites (see Fig. 5) but we have a limited number of AmeriFlux sites in California that do not cover all ecosystems.  As such, we do not report GPP here and have included an asterisk to highlight the caveats with the relationship presented here.  Future work to obtain a more robust SIF-GPP relationship covering more ecosystems is warranted."*

**Specific Comments**

**1.)**  p1, L3: "oversampling and downscaling" –> simply "downscaling" would probably be more clear for most of the readers.

We agree, updated.

**2.)**  p2, L3: "more direct methods" than what methods?

Thanks for catching this.  Updated to the following:

Page 2, Line 3: *"As such, methods of inferring..."*

**3.)** p2, L24: Several SIF downscaling methods have been published in the last years which are actually not based on machine learning nor intended to produce spatially-continuous SIF data sets from OCO-2 SIF retrievals. In particular, the method by Duveiller et al. to downscale GOME-2 SIF to 0.05° (last implementation here https://www.earth-syst-sci-data-discuss.net/essd-2019-121/) could also be adapted to TROPOMI. Please, discuss pros and cons of the oversampling/downscaling method presented in this manuscript with respect to that by Duveiller et al. and any other comparable downscaling method.

We appreciate Professor Guanter pointing out the Duveiller paper, however it is still under review at ESSDD and we prefer to cite final published papers in case there are changes during the review process. Additionally, the next line in our manuscript points readers to the review paper from Mohammed et al. (2019), a review article that is 39 pages long.

Page 2, Line 25: *"Mohammed et al. (2019) presents a detailed review of different remote sensing techniques for retrieving SIF from space-borne measurements."*

**4.)** p4, L10: "near-infrared and shortwave infrared".

Thank you for catching this, updated.

**5.)** p4, L19: "The TROPOMI SIF retrieval uses..." I don't think the average reader will understand this sentence without any further introduction to PCA-based SIF retrievals.

This line was included based on feedback from Professor Dennis Baldocchi (UC Berkeley; he provided feedback on an earlier version of the manuscript). He requested more details on the remote sensing and retrieval this was our balance between brevity and an exhaustive description: providing a few important points to those who work on SIF retrievals with references for interested readers to follow.

**6.)** p4, L 25: I can't find any information on cloud filtering, so I assume that the authors are simply not applying any. Please, discuss this here, e.g. whether no cloud filtering could/should applied globally when using SIF as a proxy for GPP (which would be somewhat scary...).

We use the same cloud filtering as Köhler *et al*. (2018). We filter pixels with VIIRS cloud fractions larger than 0.8. Text is updated as follows:

Page 4, Line 17: *"Köhler et al. (2018) filtered out pixels with cloud fractions larger than 80% based on VIIRS observations; we use this same cloud filtering here."*

**7.)** Fig 8. Panel C?

The updated manuscript now includes a Panel C showing the difference between fall 2019 and fall 2018.

**Reviewer #2 Comments:**

This work is highly interesting and its the main contribution to the scientific community is twofold: a) they introduce novel methods of oversampling and downscaling to the SIF community which offers the exciting opportunity to analyse SIF data at unprecedented spatial resolution. They also raise awareness of possible prominent retrieval biases related to bright surfaces. b) The different dynamics between canopy greenness and photosynthetic activity and the resulting benefits of SIF in general, and of TROPOMI SIF in particular, to track GPP dynamics.

However, I have two major concerns regarding the second point and main message of the paper which in my opinion are necessary to address before a publication in Biogeosciences:

We thank the Reviewer for their detailed feedback and comments on the work.

**General Comments**

**1.)** First, it is not fully clear from the work description if the comparison between greenness and SIF is meaningful: Have any filters been applied to either tower measurements, SIF or the MODIS data to remove low quality data? Also, the authors mention a clear-sky bias of the reflectance measurements as a possible explanation for the different dynamics. This implies to me that the data of VIs and SIF have not been matched in space and time before aggregating to the spatial averages shown in Fig.6. If this was indeed the case, the time series are representative of different places and therefore not fully comparable. I would like to ask the authors to clarify and if need be, to improve on this point to corroborate the main message of the paper.

Figure 2 (the scatterplot comparison) is a direct comparison between MODIS observations at the same location on the same day. This figure serves as a one-to-one comparison of the different products and the version in the supplement is expanded to include comparison with the downscaled SIF products.

For Figure 6, we interpolate the MODIS data in time for gap-filling purposes and then compute a statewide mean. The same number of points go into the statewide mean for both SIF and MODIS but there is more gap filling for MODIS. The statewide mean SIF and VIs represent our best attempt at producing a statewide mean for the different variables. The MODIS data will have gaps that need to be filled via interpolation during cloudy scenes, leaving only clear-sky conditions to build a statewide mean. TROPOMI will observe more scenes with low-to-moderate cloud cover, thus potentially inducing a clear-sky bias in the inferred statewide seasonal cycle. So the point is that the inferred seasonal cycle is different when using SIF vs VIs and part of that difference is likely due to the lack of data in cloudy periods from MODIS.

Figure 2 Caption: *"Panels show a comparison of coincident measurements in both space and time."*

**2.)** Second, I see the explanation for the different dynamics in SIF and greenness as incomplete. The authors convincingly argue that the different phasing of activity between evergreen forests and grasses, chaparral, and oak savanna causes the double peak in SIF. However, a similar decomposition by land cover type (Fig.6a) is missing for the greenness indices and I strongly suggest to include this in the analysis (at least for one of the indices) in order to get an idea of where/ in which ecosystems SIF and greenness are particularly dissimilar (Fig.6c). Otherwise, a sentence like in the abstract that SIF 'can detect the downregulation of photosynthesis even when plants appear green' is not justified from the material presented in the paper. Finally, this analysis of where and when VIs and SIF disagree, could be completed by a driver analysis to understand which processes does SIF see that greenness does not and to undermine the argumentation in p.17 ll.15-28. There are features as those in May in both years, which coincide with similar dips in light and rain events, it is not clear which of these is more important for which ecosystem. There are other prominent features such as the smaller peak in September 2018 in SIF which does not seem to have an obvious relationship with either precipitation or light.

The line from the abstract was based on inferences from previous work and has been amended to indicate this. See, also, the response to Comment #1 from Professor Guanter. We have included the requested driver analysis for MODIS NIR$_v$ as two additional supplemental figures (see below).

Page 1, Line 9: *"The different seasonality in the vegetation indices may be due to a clear-sky bias in the vegetation indices, whereas previous work has shown SIF to have a low sensitivity to clouds and to detect the downregulation of photosynthesis even when plants appear green."*

[Figure]

Figure S6: Same as main text Fig. 6 but for MODIS NIR$_v$.

[Figure]

Figure S7: Same as main text Fig. 8 but for MODIS NIR$_v$.

**Minor Comments**

**1.)** The higher correlation between greenness and SIF at longer time scales is mentioned both in the abstract and conclusions but in the main text only in a sub-clause, and is not a main finding of your work. I see this as distracting side information as well, which does not necessarily need to be mentioned in both the abstract and the conclusions.

We disagree with the reviewer on this point. Much of the previous work comparing NIR$_v$ and SIF was done at monthly or annual timescales and found stronger relationships (e.g., Badgely *et al.*, Science Advances 2017). As such, we do feel that it is important but there is not much additional explanation needed to understand this.

**2.)** Abstract: *"The different seasonality in the vegetation indices may be due to a clear-sky bias in the vegetation indices, whereas SIF has a low sensitivity to clouds and can detect the downregulation of photosynthesis even when plants appear green."* This sentence illustrates my major comment from above that the question of what drives the SIF response in the different ecosystems is not sufficiently covered by the analysis.

Updated to reflect that the latter inference was based on previous work. See response to Comment #1 from Professor Guanter and Comment #2 from Reviewer #2.

Page 1, Line 9: *"The different seasonality in the vegetation indices may be due to a clear-sky bias in the vegetation indices, whereas previous work has shown SIF to have a low sensitivity to clouds and to detect the downregulation of photosynthesis even when plants appear green."*

**3.)** The fact that there is a double peak in SIF but not in the VIs is mentioned twice in the conclusions.

Updated.

**4.)** Apart from the fact, that the scaling from SIF to GPP is not needed in this manuscript, it is rather uncommon to use the unit of mu mol/m$^2$/s from the tower measurements also for seasonal values as in the maps in Fig 6. gC/m$^2$/day is rather common.

The AmeriFlux data are provided in units of μmol/m$^2$/s and a number of papers use these same units. For example, the paper describing FLUXNET (Baldocchi *et al.*, 2001) uses these units for some of their figures. Magney *et al.* PNAS (2019) also used these units for some of their figures. Further, these units are useful for my own work with $CO_2$ flux inversions.

**5.)** Fig 6G' does not exist, pay attention in caption and Fig 6b.

The updated manuscript now includes Fig. 6G'.

**6.)** p.9 l.34: Can you really resolve daily features with an average over 14 days despite daily sampling?

We thank the review for pointing this out. We did not mean to imply that we resolve daily features, it's intended to highlight that we are **producing** our estimate every day (based on a 14-day window). Text has been updated to highlight this:

Page 9, Line 27: *"…Köhler et al. (2018) seasonal cycle is produced at weekly temporal frequency whereas we produce daily estimates using a 14-day moving window."*

**7.)** p.11 l.29 -p.12 l.2: To my (admittedly non-native English) ears the word 'owing' in this sentence sounds misplaced.

This is grammatically correct.

**8.)** Fig 6B: why is the cropland contribution stressed in this panel?

Cropland is highlighted because this is how we chose the time period for Panel G and G'.

**9.)** p.19 l.26: *"it seems unlikely that the grasslands and forests will exhibit opposing responses to a forcing."* It probably depends and an extended analysis as suggested above can give indications of whether this is true for California or not. There are counter examples e.g. in Flach *et al.* 2018 https://doi.org/10.5194/bg-15-6067-2018 or Walther *et al.* 2019 https://doi.org/10.1029/2018GL080535

There could certainly be a contrasting response, but it seems more likely that it falls out of the EOF requirement to compactly represent the system.  From the abstract of Mohanan *et al*. (2009), a review paper on EOFs:

> "Often in the literature, EOF modes are interpreted individually, independent of other modes. In fact, it can be shown that no such attribution can generally be made. This review demonstrates that in general individual EOF modes (i) **will not correspond to individual dynamical modes**, (ii) **will not correspond to individual kinematic degrees of freedom**, (iii) **will not be statistically independent of other EOF modes**, and (iv) **will be strongly influenced by the nonlocal requirement that modes maximize variance over the entire domain**. The goal of this review is not to argue against the use of EOF analysis in meteorology and oceanography; rather, it is to demonstrate the care that must be taken in the interpretation of individual modes in order to distinguish the medium from the message."
* * *
**Short Comment from Mr. Paolo Tasseron & Prof. Wouter Peters:**

This review was prepared as part of graduate program course work at Wageningen University, and has been produced under supervision of Prof Wouter Peters. The review has been posted because of its good quality, and likely usefulness to the authors and editor. This review was not solicited by the journal.
…
In my opinion, the study is interesting and introduces a relevant novelty in the narrow scientific community bridging remote sensing science and photosynthesis research. However, I think three flaws are present in the current manuscript, of which I recommend some revisions before publication.

We thank Mr. Tasseron and Professor Peters for their feedback on the work.  We have responded to their comments below.

**General Comments**

**1.)**  To start with the first issue, on page 9 (lines 7-27) a fourteen-day moving window is used in combination with a spatial downscaling method to obtain daily estimates of SIF at a high resolution. In combination with the consequent pre-weighting of the SIF signal by the underlying vegetation fraction (MODIS $NIR_v$), large-scale changes in spatio-temporal patterns are conserved. On lines 20-21, the authors assume that the observed SIF from TROPOMI likely originates from more vegetated regions within that scene. However, the $R^2$ value of the linear correlation between SIF and $NIR_v$ (0.52, Figure 2 on page 7) implies that a significant part of the variance in SIF cannot be explained by the underlying vegetation fraction. Besides, by using the averaged value of the 14-day moving window, a pseudo-daily average SIF value is created, rather than the actual daily value. This is fine, provided that a certain accuracy assessment is conducted. Especially because the authors mention, on page 9 lines 31-32, significant differences are found with the similar method of Köhler et al. (2018) in which a quality control and accuracy assessment are indeed present. In addition, the downscaling (from 24.5 km$^2$ to 0.25 km$^2$) is likely to introduce inaccuracies, which requires quantification.

There is an implicit physiological argument being made here. Solar induced chlorophyll fluorescence is *inherently* a signal emitted from chlorophyll. As such, one would expect that the measured SIF signal to originate from regions within a TROPOMI scene that have more vegetation and chlorophyll.

For the second part of this comment (*"the $r^2$ of the linear correlation between SIF and $NIR_v$ implies a significant part of the variance in SIF cannot be explained by underlying vegetation fraction"*), the $r^2$ Mr. Tasseron refers to is using two years of data over the entire state. So it implies that SIF and $NIR_v$ do observe similar (but not identical) phenomena, making it an excellent candidate for downscaling. Regarding Mr. Tasseron's criticism of our comparison with the Köhler *et al.* (2018) paper: differences are found at fine spatial scales but large-scale patterns are consistent. Finally, we point Mr. Tasseron to our response to Comment #1 from Reviewer 2 on the temporal differences.

**2.)** Secondly, from page 11 onwards, the authors use a method to infer GPP from SIF, based on light-use efficiency and the probability of SIF photons escaping the canopy. Interestingly, Paul-Limognes *et al.* (2018) found that SIF was more affected by environmental conditions than GPP. Midday-depressions in SIF were linked to peak VPD values following peak photosynthetic photon flux density (PPFD). Besides, Walther *et al.* (2016) state that in evergreen needle-leaf forests, the length of the photosynthetically active period indicated by SIF is up to six weeks longer and commences a month earlier than the green biomass changing period proxied by EVI. Even though the authors used $NIR_v$ instead of EVI to downscale SIF, the different timing could significantly alter the double peak structure. Moreover, the authors state there is a lack of GPP measurements in evergreen forests, while much of California is dominated by this vegetation type (page 13, line 17-19). In combination with the a-synchronous SIF/MODIS dynamics, this will propagate into a major bias in the scaling factor of 18.5 ± 4.9 which is inferred on page 13, line 14. Therefore, I think that the equation on page 13, line 20 (*GPP := 18.5 • SIF*) should include a revised quantification of the error margins. In doing so, the authors should determine an alternative error margin whilst taking into account the fractional contribution of evergreen forests to GPP. The latter can best be inferred from a biosphere model or studies which used eddy-covariance measures in similar evergreen forests.

Regarding the first point that the use of MODIS vegetation indices could impact the double peak, it does not. The large-scale patterns are invariant to the choice of oversampling or downscaling. This can be clearly seen in the inset in the left column of Figure 4 where both the oversampling and downscaling result in the same 2018 seasonal cycle. The major features (i.e. the double peak) are also present in the Köhler *et al*. (2018) gridding for California. This is because the oversampling and downscaling conserve the SIF over a given TROPOMI pixel, so averaging to coarser spatial scales will yield an identical seasonal cycle. For the second point, see our response to Comment #2 from Professor Guanter.

The following text has been added:

Page 13, Line 23: *"To reiterate, there is a clear correspondence between the observed SIF and GPP estimated for the different AmeriFlux sites (see Fig. 5) but we have a limited number of AmeriFlux sites in California that do not cover all ecosystems. As such, we do not report GPP here and have included an asterisk to highlight the caveats with the relationship presented here. Future work to obtain a more robust SIF-GPP relationship covering more ecosystems is warranted."*

**3.)** Lastly, the authors successfully identify a double-peak in the seasonality of GPP. However, the number of (recent) references concerning underlying reasons for this double peak or other case studies in which a double peak is found, is unsatisfactory. References to Xu & Baldocchi (2003), Xu *et al.* (2004), Xu & Baldocci (2004) explain changes in carbon fluxes between ecosystems and vegetation types well, yet the link with SIF dynamics is lacking (Page 17, lines 15-22). Perhaps the following is a cause of the state-of-the-art novelty of this subject, but there are zero references made to any other recent papers discovering the double peak in GPP/SIF. Given the importance of this conclusion to the subject of the manuscript, I highly suggest investigating and mentioning recent existing literature explaining the double peak phenomenon. If the latter turns out to be infeasible because it is such a novelty, it is suggested to emphasize the scientific novelty in this paper. For instance, Li *et al.* (2014) imply that MODIS EVI is unsuitable for detecting a double peak in vegetated areas which usually manifest double peaks. This would strengthen the relevance as to why SIF needs to be used.

We do not reference other papers on this double peak because (to our knowledge) this is the first time it has been noted. This inference could have probably been made in earlier work (e.g., the papers we cite from the Baldocchi group) but, to our knowledge, it has not been investigated before. Satellite measurements of SIF are a fairly novel measurement (first global retrievals were made in 2011) and previous work using other satellites (e.g., GOME-2) has been limited to very coarse spatial resolutions. Our work is one of the first to get down to this spatial resolution that allows separating the processes driving this.

Given this, we chose the title of the paper to highlight the novelty of the finding. We also devoted two full sections of the text to discussing the processes driving this phenomenon.

**Specific Comments**

**1.)** In Table 1, all vegetation types have two or more study sites except for the WSA (Woody Savannas). I would like to give the authors awareness that one study site might not be representative for the entire ecosystems, especially when all other vegetation types have multiple sites.

Agreed. However, there is not much we can do in response because there simply are not additional AmeriFlux sites in California. This is, again, why we include the asterisk on GPP. We have plans to extend this analysis globally to include more sites, but the focus of this study was on California.

**2.)** In Figure 1, Page 3: The description mentioned that black stars show the location of six Ameriflux sites, However I can only discern three and they seem to be closely packed at this resolution.

Many of the AmeriFlux sites are in close proximity to each other. For example, the US-Tw1 US-Tw3, US-Tw4, and US-Tw5 are located on the same island in the Sacramento Delta (their longitudes differ by less than 0.01° longitude). See below for a zoomed in view of the Sacramento Delta. So there are indeed 6 AmeriFlux sites plotted, some of the stars just lie on top of each other in the Figure. This is an important point for the comparison of the wetland sites, as there is still local heterogeneity observed at these eddy flux sites that sub-grid scale.

[Figure]

**3.)** In Figure 2 on page 7, the axes lack titles. This is relevant to include for the x-axes of the bottom row of graphs, as the range of the axes are different.

Axes cover the dynamic range for each product and units are included in the caption.

**4.)** In Figure 3 on page 8 the swath resolution is 4.0 km × 7.0 km, whereas in the text on Page 9, line 4 it is stated that this resolution is 3.5 km × 7.0 km. This should match.

The left panel is a schematic. The TROPOMI resolution at nadir is $3.5 \times 7$ km$^2$, but is larger at the edges. Supplemental Figure 1 from Köhler *et al.* (2018; see below) shows how the pixel size varies across the swath. Further, the TROPOMI team reduced the along-track integration time in August 2019 thus reducing the along-track pixel size from 7 km to 5 km. Again, this left panel referred to by Mr. Tasseron is a schematic meant to illustrate how differences in viewing geometry allow us to bisect subdivide pixels from the nominal resolution.

[Figure]

Figure S1: Across track pixel size/ground pixel area as function of spatial row/viewing zenith angle computed from soundings at the equator (same orbit as in Fig. 1).

**5.)** On page 9, line 19-20: perhaps it is necessary to introduce that the NIR$_v$ was used in the pre-weighting of SIF, rather than introducing it later on Page 11, line 5.

The pre-weighting can be applied with any vegetation index, the rest of the paper simply uses NIR$_v$ because it showed the strongest correspondence with SIF. Supplemental Figure 2 actually shows a comparison of the SIF downscaled using other MODIS vegetation indices (NDVI and EVI) as well. So we prefer to keep this expression more general here.

**6.)** On page 14 in the figure description, a reference to Panel G' is made, whereas this panel is not present in the accompanying figure (6).

Updated to include Panel G'.

**7.)** On page 16, line 8-9 it is stated that a 'reasonable consistency' is found. This should be quantified.

The figure is the quantification of the difference between the years.

**8.)** In the conclusion on page 20, parts of line 6-7 and line 22-23 have very similar information.

Updated.

[revised manuscript text omitted]

---

## Author Response (AR2)

**Response to Comments:**

Your revised manuscript has now been seen by two reviewers and both are satisfied with your revisions. One reviewer did request that you include some short information on data filtering and interpolation (as given in the responses to reviewers) in the main text for reproducibility reasons. This seems like a good addition to me, would this be possible? In addition,

We have now included an additional supplemental section (Section S2) with this information. We agree that it is important to include but it seems overly detailed for the casual reader. As such, the supplement seems like a good place for this information so an interested reader can in exactly reproduce that figure.

Supplemental Section S2: *"Main text Figure 2 (the scatterplot comparison) is a direct comparison between MODIS observations at the same location on the same day. This figure serves as a one-to-one comparison of the different products and supplemental Figure S2 is expanded to include comparison with the downscaled SIF products. For main text Figure 6, we interpolate the MODIS data in time for gap-filling purposes and then compute a statewide mean. The same number of points go into the statewide mean for both SIF and MODIS but there is more gap filling for MODIS. The statewide mean SIF and VIs represent our best attempt at producing a statewide mean for the different variables. The MODIS data will have gaps that need to be filled via interpolation during cloudy scenes, leaving only clear-sky conditions to build a statewide mean. TROPOMI will observe more scenes with low-to-moderate cloud cover, thus potentially inducing a clear-sky bias in the inferred statewide seasonal cycle. As such, the inferred seasonal cycle is different when using SIF vs VIs and part of that difference is likely due to the lack of data in cloudy periods from MODIS."*

-The caption of Fig.4: you have 'but' twice
-p.17 l.34: two transects, not three

We thank the reviewer for pointing these out and have corrected them.

[revised manuscript text omitted]